# Long-Term (1986–2015) Crop Water Use Characterization over the Upper Rio Grande Basin of United States and Mexico Using Landsat-Based Evapotranspiration

**Gabriel B. Senay** [1,]*, **Matthew Schauer** [2], **Naga M. Velpuri** [3], **Ramesh K. Singh** [3], **Stefanie Kagone** [3], **MacKenzie Friedrichs** [4], **Marcy E. Litvak** [5] **and Kyle R. Douglas-Mankin** [6,†]

1   U.S. Geological Survey (USGS), Earth Resources Observation and Science (EROS) Center, North Central Climate Science Center, Fort Collins, CO 80523, USA
2   Innovate! Inc., Contractor to the U.S. Geological Survey EROS Center, Sioux Falls, SD 57198, USA
3   ASRC Federal Data Solutions, Contractor to the U.S. Geological Survey EROS Center, Sioux Falls, SD 57198, USA
4   KBR, Contractor to the U.S. Geological Survey EROS Center, Sioux Falls, SD 57198, USA
5   Biology Department, University of New Mexico, Albuquerque, NM 87131, USA
6   USGS New Mexico Water Science Center, Albuquerque, NM 87113, USA
*   Correspondence: senay@usgs.gov
†   Current address: U.S. Department of Agriculture, Agricultural Research Service, Fort Collins, CO 80526, USA.

**Abstract:** The evaluation of historical water use in the Upper Rio Grande Basin (URGB), United States and Mexico, using Landsat-derived actual evapotranspiration (*ETa*) from 1986 to 2015 is presented here as the first study of its kind to apply satellite observations to quantify long-term, basin-wide crop consumptive use in a large basin. The rich archive of Landsat imagery combined with the Operational Simplified Surface Energy Balance (SSEBop) model was used to estimate and map *ETa* across the basin and over irrigated fields for historical characterization of water-use dynamics. Monthly *ETa* estimates were evaluated using six eddy-covariance (EC) flux towers showing strong correspondence ($r^2 > 0.80$) with reasonable error rates (root mean square error between 6 and 19 mm/month). Detailed spatiotemporal analysis using peak growing season (June–August) *ETa* over irrigated areas revealed declining regional crop water-use patterns throughout the basin, a trend reinforced through comparisons with gridded *ETa* from the Max Planck Institute (MPI). The interrelationships among seven agro-hydroclimatic variables (*ETa*, Normalized Difference Vegetation Index (NDVI), land surface temperature (LST), maximum air temperature (*Ta*), potential ET (*ETo*), precipitation, and runoff) are all summarized to support the assessment and context of historical water-use dynamics over 30 years in the URGB.

**Keywords:** evapotranspiration; remote sensing; SSEBop model; water-use trends; Landsat; Upper Rio Grande Basin

## 1. Introduction

Actual evapotranspiration (*ETa*) is the hydrologic process that converts liquid water on the soil-vegetative surface into atmospheric water vapor. Evapotranspiration (ET) includes the evaporation from the surface of plants and soil as well as the transpiration of water as vapor through plant stomata. At a basin level, *ETa* is a key component of the hydrologic cycle and water budget, recycling 60–75% of total terrestrial precipitation [1,2]. In the United States, the ratio of *ETa* to precipitation ranges from less

than 20% in the Northwest, which is dominated by high rainfall and moderate temperatures, to over 80% in the arid Southwest, which is characterized by low rainfall and high temperatures [3]. Due to the large range in *ETa* variability and the close association of *ETa* with soil moisture, vegetative health, and atmospheric demand, *ETa* can serve as a valuable indicator of landscape response to environmental and climatic drivers.

The process of recycling water into the atmosphere via ET involves not only the transport of water, but the transfer of energy as evaporative cooling also reduces the temperature of the land surface. Thus, *ETa* can be estimated using either a mass balance or an energy balance approach. The mass balance approach requires the tracking of supply (rainfall and/or irrigation) using prognostic modeling where *ETa* is determined as a function of soil moisture and vegetative condition for a given atmospheric demand [4,5]. An energy balance approach, on the other hand, relies on land surface temperature (LST) to measure evaporative cooling in the soil-vegetation-atmosphere system—a diagnostic modeling approach to estimating *ETa* [6].

Advances in remote sensing, gridded weather datasets, and process-based algorithms can produce spatially explicit *ETa* maps across large areas, whereas direct measurements are limited to select locations using expensive instrumentation. Most remote sensing-based ET models employ key variables such as LST, as in Operational Simplified Surface Energy Balance (SSEBop) or SEBAL (Surface Energy Balance Algorithm), METRIC (Mapping Evapotranspiration at high Resolution with Internalized Calibration), ALEXI (Atmospheric-Land Exchange Inverse), or Normalized Difference Vegetation Index (NDVI), such as in the SIMS method (Satellite Irrigation Management System), in approaches that range from empirical to full energy balance algorithms [7–18]. A comprehensive review of the different remote sensing methods for ET modeling is summarized by various researchers [19–23].

As the archive of data expands, satellite data are being used for trend studies in *ETa*. As opposed to seasonal drought monitoring and spatial assessment of *ETa* by land cover type, trend analysis faces more challenges from consistency in data quality and model stability across years. Most studies that have looked at trends in *ETa* have relied on the combination of climatic datasets such as potential evapotranspiration and NDVI for model parameterization at regional and global scales using satellite sensors such as Advanced Very High Resolution Radiometer (AVHRR, since 1970s) and Moderate Resolution Imaging Spectroradiometer (MODIS, since 2000) for characterizing *ETa* for various land cover types [24,25]. In addition to NDVI, Jin et al. [26] examined recent (2001 to 2014) trends in *ETa* at a regional scale using the Surface Energy Balance System (SEBS) model.

*ETa* trend analysis over croplands are more common with the use of the crop coefficient (Kc) approach where NDVI is used to estimate Kc, which in turn is used to scale atmospheric demand such as reference ET to study field-based water-use trends of specific crops such as corn and soy [27–29]. Other techniques include the use of the complementary *ETa* relationship, which has been applied to investigate long-term (1979–2015) crop *ETa* trends at a watershed scale [30].

Crop coefficient-based historical *ETa* trends are important in determining crop water requirements in well-managed irrigated agriculture that is operating under optimum agronomic practices and water management. In actual practices, Samani et al. [31] list several reasons for the variability of field-level *ETa* and crop coefficients, including limited water supply, knowledge of irrigation scheduling, cultural practices, and economic factors among others.

The use of Landsat data for quantifying historical crop water-use dynamics and trend analysis at a regional scale has not been reported. However, Landsat data have been used to study *ETa* trends (1975–2011) over wetland vegetation through the scaling of potential ET with NDVI [32]. The foundation of this study is based on several of our previously published research efforts. Singh et al. [33] and Senay et al. [34] demonstrated the use of Landsat thermal data to characterize the spatial and seasonal distribution of crop water use for the Colorado River Basin for individual years. Furthermore, Senay et al. [35] reported on the importance of land and water management policy changes in driving *ETa* trends in the Palo Verde irrigation district, California, during 1984–2014 using thermal-based ET modeling.

Several studies have reported on the estimation and mapping of *ETa* in the western United States and the Upper Rio Grande Basin (URGB) using remote sensing techniques for natural systems and quantifying *ETa* by crop types [14,36–39]. However, no study has reported on *ETa* for the entire URGB over 30 years at the Landsat scale to characterize the spatiotemporal dynamics of *ETa* including trend analysis. This study applies comparable modeling techniques with improved model parameterization—from previous studies in the Colorado River Basin and in California—on the URGB over 30 years using Landsat datasets and weather parameters [33–35].

The overall goal of this study was to estimate and characterize historical water use in the URGB using Landsat-derived evapotranspiration (*ETa*) from 1986–2015 using the SSEBop *ETa* model. The specific objectives were to (1) evaluate accuracy of SSEBop *ETa* using in situ observations from six eddy-covariance flux towers (2007–2014) in the study region, (2) evaluate long-term trend results using historical (1986–2011) gridded FLUXNET datasets, and (3) quantify spatiotemporal distribution of crop water use in the basin (1986–2015) at basin and region scales.

In this study, 10,335 Landsat images (Landsat 5, 7, and 8) were processed using the SSEBop model to quantify water use for the URGB for 30 years. This is the first *ETa* study to be completed at this spatial and temporal scale. We present the spatiotemporal dynamics of *ETa* at the scale of the basin, state regions, and pixel-level including seasonality and historical water-use trends. The interrelationships among seven agro-hydroclimatic variables (*ETa*, NDVI, LST, maximum air temperature (*Ta*), maximum potential evapotranspiration (*ETo)*, precipitation, and runoff) are summarized to contextualize the historical water-use dynamics in the URGB.

## 2. Methods

### 2.1. Upper Rio Grande Basin (URGB)

The Rio Grande is the fifth longest river in North America and extends over ~960 km from its headwaters in the southern Colorado Rocky Mountains through the south-central United States and Mexico, forming part of the international border between the two countries, and ending at the Gulf of Mexico [40]. The primary source of surface water in the northern part of the basin is due to spring runoff (snowmelt) from the southern Rocky Mountains, and the southern reach of the river flows through the Chihuahuan Desert where annual precipitation averages 200 mm, most of which occurs in the summer monsoon season [40]. Like many arid regions, there are competing demands for a limited water supply and agricultural areas are prone to drought, which negatively impacts many water users within the basin. Much of the late 1990s and early 2000s saw prolonged drought conditions within the URGB where basin inflows reached only 10% of the 30-year average; in 2004, water storage in the Elephant Butte Reservoir, the largest reservoir in the basin, dropped to less than 6% of its total capacity [40]. In the URGB, like much of the southwestern United States, 80–90% of the river flow is used for irrigated agriculture, but only a portion, ranging from 30–70%, is used consumptively; the remainder recharges the groundwater, supplies riparian habitats, or returns to river flow [40].

The URGB extends across Colorado, New Mexico, and Texas in the United States and across the State of Chihuahua in Mexico. (Figure 1). There are multiple major irrigation districts within the URGB including the San Luis Valley (SLV) in south-central Colorado, the Middle Rio Grande Conservancy District (MRGCD) near Albuquerque, New Mexico, the Elephant Butte Irrigation District (EBID) near Las Cruces, New Mexico, and the El Paso County Water Improvement District near El Paso, Texas. For the purposes of this study, *ETa* trends were sampled for all irrigated cropland areas within the URGB.

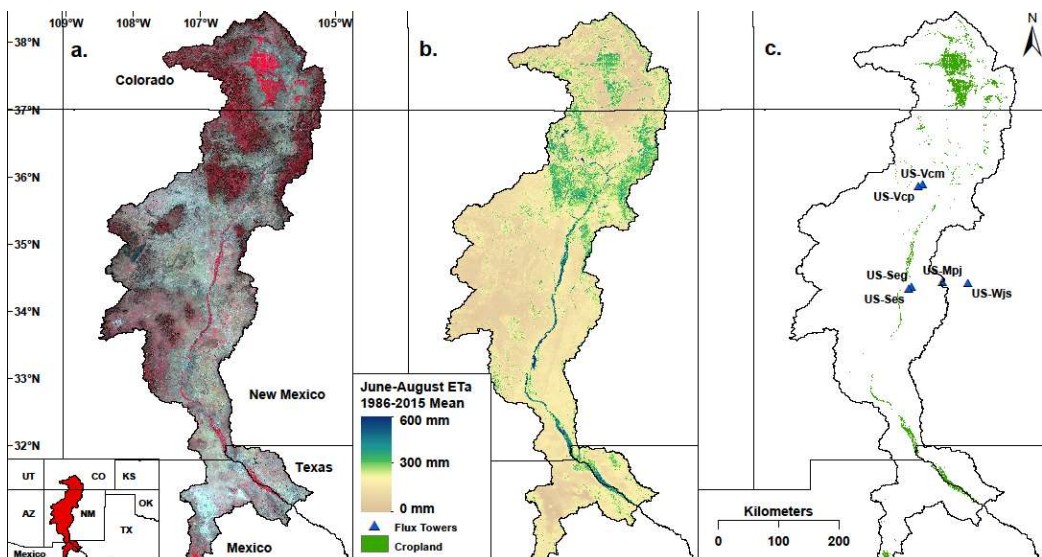

**Figure 1.** The Upper Rio Grande Basin including (**a**) false color composite (near infra-red, red, and green) of Landsat based on greenest-pixel (NDVI) from 2010–2015 with dense vegetation shown in red; (**b**) 1986–2015 mean Operational Simplified Surface Energy Balance (SSEBop) actual evapotranspiration (*ETa*) from June–August (summer), and (**c**) maximum cropland extent for the basin with locations of the six eddy-covariance flux towers.

The Colorado section of irrigated areas in the URGB is mostly composed of the SLV and its associated irrigation districts. The SLV is a high-altitude agricultural valley, an alpine desert, in south-central Colorado at an elevation of approximately 2300 m (7600 ft) and bordered by the San Juan Mountains to the west and the Sangre de Cristo Mountains to the east [41]. The valley averages less than 200 mm of annual precipitation as it lies in the rain shadow of the San Juan Mountains, which receive 1000–2000 mm of precipitation annually [41]. The hydrologic system of the SLV is primarily driven from snowmelt, with most of the water coming from river and stream flow from the surrounding mountain ranges [42]. Due to the high elevation and latitude, the SLV experiences a shorter growing season and colder winters (winter low temperatures of less than 253 K or −20 °C) than the southern regions of the URGB [41,43]. Despite the arid climate, the valley is a major agricultural region for Colorado, producing much of the state's potato (14,000–30,000 hectares (ha)) and alfalfa (80,000–100,000 ha) crops. Agriculture in SLV accounts for 30% of the valley economy [41]. The arid conditions require virtually all agriculture to be irrigated and reliant on surface water of the Rio Grande and groundwater reserves in the valley's aquifer [41].

Much of the irrigation in New Mexico is clustered around the Rio Grande, which flows south through the URGB. The Rio Grande is bordered by bosque, or riverside forest, but within the narrow floodplain of the river, there is widespread irrigated agriculture despite receiving less than 250 mm of annual rainfall [44]. In New Mexico, irrigated agriculture is dependent on surface water provided by administrative irrigation districts such as the MRGCD or the EBID with some supplemental groundwater from privately owned wells [45]. The largest concentration of irrigated agriculture in the basin is in the Rincon and Mesilla Valleys, which extend from Elephant Butte Reservoir, through Las Cruces, New Mexico, and end at the New Mexico/Texas border.

About 80% of irrigated lands in the EBID are devoted to the production of alfalfa, cotton, and pecans [45]. Pecan production in the EBID accounts for a large proportion of the district's consumptive water use with 11,625 ha of pecan trees (*Carya illinoinensis*) according to the 2012 U.S. Census of Agriculture [46,47]. The "thirsty" label for pecan was investigated by Skaggs et al. [46] using lysimeters and evaporation pan techniques, which found that the seasonal consumptive use (April–October) of

mature pecan trees near Las Cruces ranged from 1000–1300 mm for closely spaced, full-grown trees; however, these were limited-scope, site-specific studies.

### 2.1.1. Landsat Imagery

Pre-Collection 1 Landsat imagery was collected for 13 path/rows (path 32/rows 38–39, path 33/rows 34–39, path 34/rows 33–37) from scenes between 1986 and 2015 with less than 60% cloud cover (https://www.usgs.gov/centers/eros). In total, 10,335 Landsat images were collected from all Landsat satellites (5/7/8) with 54% of the images coming from Landsat 5 (5614), 39% from Landsat 7 (4019), and a further 7% from Landsat 8 (702) as presented in Figure 2. For years with only one satellite in operation, specifically 1984–1998 (Landsat 5) and 2012 (Landsat 7), the average number of path/row images was 14 per year (Figure 2). Previous studies have suggested at least 10–12 images per year are desired to generate a reliable *ETa* estimate [33]. With two satellites in operation (1999–2011; 2013–2015), the average number of Landsat images increased to 36 per year in a path/row. Roughly half of this study (16 of 30 years) featured a single satellite in operation, but the model was applied similarly for the entire period regardless of the number of satellites in operation as there was still enough imagery to provide reliable *ETa* estimates. As there were fewer than 10 images per year for 1984 and 1985, we omitted these two years from our analysis.

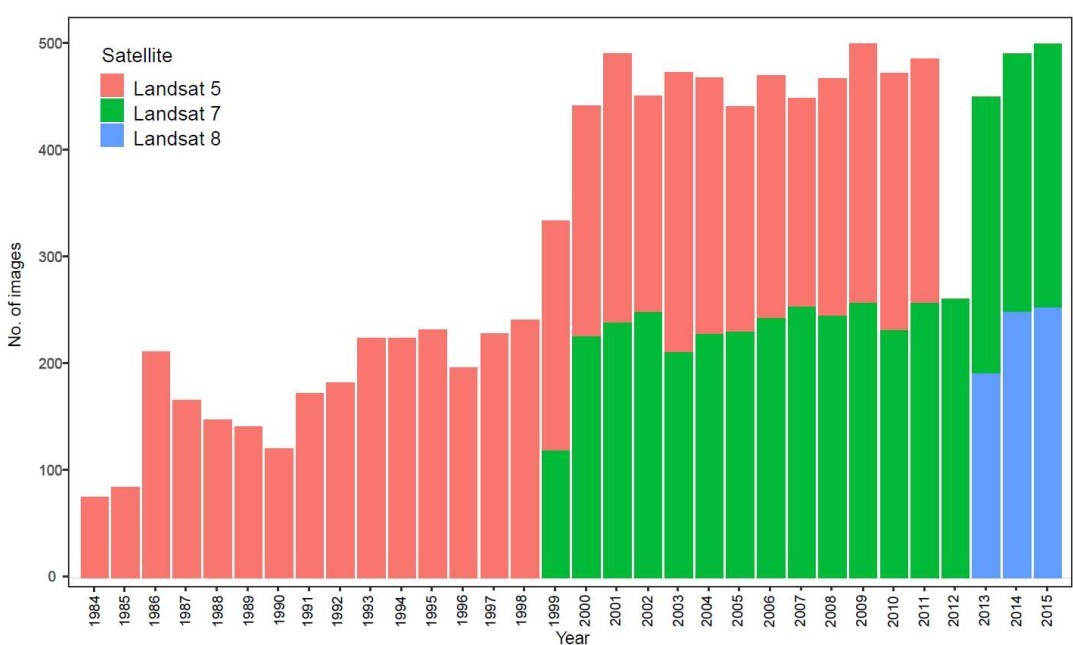

**Figure 2.** Number of Landsat images (<60% cloud cover) available for the entire Upper Rio Grande Basin (URGB) from 1984–2015. This study used images from the 1986–2015 period.

### 2.1.2. Hydro-Climatic and Model Parameter Datasets

Other model inputs included *Ta* from TopoWX (http://www.scrimhub.org/resources/topowx), grass reference *ETo* from GridMET (http://www.climatologylab.org/gridmet.html), albedo by the Land Processes Distributed Active Archive Center (LP DAAC, https://lpdaac.usgs.gov/), and temperature differential (*dT*) generated from the SSEBop model [33,48,49]. TopoWX generates the *Ta* dataset at 800-m spatial resolution, provided on a yearly basis for 1948–2016. The *ETo* dataset from GridMET, like TopoWX *Ta*, is for the conterminous United States (CONUS-wide), provided daily for each year but at 4-km resolution from 1980 to the present. The Albedo dataset is a pre-computed daily median derived from the White Sky Albedo provided by the LP DAAC at 1-km spatial resolution and is based on 10 years of data (2001–2010). The parameter *dT* is a pre-defined daily temperature difference between bare-dry-surface and canopy-level air temperature [6,33], which is unique for each pixel and

day-of-year, calculated under clear-sky conditions for a bare, dry soil where *ETa* is assumed to be zero and sensible heat is assumed to be maximum, which includes advective energy.

### 2.1.3. Preprocessing

Once the Landsat images were collected, clouds and cloud shadows were removed using the FMask (Function of Mask) algorithm [50]. However, in some images, we found undetected clouds and cloud shadows that were missed by the FMask algorithm. To further remove these 'cloud contaminated' pixels, a secondary cloud buffer was applied using a masking threshold greater than 15 degrees (K) difference between the *Ta* (maximum air temperature) and *Ts* (Landsat LST), known as *Tdiff*, to remove unrealistic cold outliers. After removing clouds and cloud shadows, NDVI was calculated using top-of-atmosphere reflectance bands (red and near-infrared), and *Ts* was calculated from the thermal band aided by emissivity values derived from NDVI [34].

### 2.2. SSEBop Model

The SSEBop model calculates *ETa* from each Landsat image using an algorithm that takes *Ts* and *ETo* as primary model forcing inputs [6,16]. SSEBop is an energy balance-based approach but only solves for latent heat flux at the daily time scale using a satellite psychrometric approach to determine a daily index or ET fraction (*ETf*) for every Landsat pixel, which, when used with *ETo*, produces a daily *ETa* estimate [6]. The *ETf* is calculated per-pixel using Equation (1):

$$ETf = 1 - \gamma^s(Ts - Tc) \tag{1}$$

where *ETf* is the daily ET fraction for each pixel nominally ranging between 0 and 1; $\gamma^s$ is the 'surface' psychrometric constant over a dry-bare surface, and it is the same as the inverse of the *dT* parameter in Senay et al. [16]. *Ts* (K) is derived from the Landsat thermal band; and *Tc* is the coldest/wettest surface temperature (K) limit, derived from *Ta* [6,35]. The constant 1 represents the ET fraction value during maximum ET, i.e., when *Ts* = *Tc*. Daily *ETa* is then determined on a per-pixel basis using Equation (2):

$$ETa = ETf * k * ETo \tag{2}$$

where *ETa* is actual ET (mm); *ETo* is (grass-reference) potential ET (mm); and *k* is a scaling coefficient of 1.25, which scales-up the grass-reference *ETo* to an alfalfa-reference type.

Before *ETa* is computed, cloud-masked pixels in each ET fraction image were filled using per-pixel linear interpolation from temporally adjacent *ETf* images ranging up to 48 days before and after the image date. The interpolation method is more effective with multiple images, but in cases where there were insufficient images to complete the interpolation algorithm, a monthly median ET fraction, taken from all years, was used to fill missing pixels. Monthly total *ETa* was calculated by aggregating daily *ETo* into equal periods and then using Equation (2) with the corresponding *ETf* grid for that period. These monthly *ETa* totals are then mosaicked together for all path/rows using a mean operator in scene-overlap areas. Monthly SSEBop ETa data for URGB are available as a supplementary material at Available online: https://earlywarning.usgs.gov/ssebop/landsat/605 (accessed 3 July 2019).

For this study, parameters listed in Table 1 were implemented. These parameters are slightly different from those reported in Senay et al. [35] based on updated information on model performance, but the overall change is rather small at a large spatial scale. Daily median *Ta* grids were computed using *Ta* from 1984–2015 (32 years) provided by TopoWX [49]. The SSEBop c factor, which converts *Ta* into *Tc*, was applied per Senay et al. [35] with the median *Ta* dataset rather than yearly *Ta* data for the purposes of consistency and simplicity. In addition, improved *Tdiff* thresholds used in cloud masking as described in Section 2.1.3 were identified and included for all calibration and scene processing steps.

**Table 1.** SSEBop model parameter description and constraint limits.

| Parameter | Constraint Limits | Outcome |
|---|---|---|
| c factor | NDVI > 0.7<br>$T_s$ > 270 K<br>$0 <= (T_a - T_s) <= 10$ K<br>Mean of $(T_s/T_a) - 2$ STD | c factor based on coldest and greenest vegetation (5th percentile) for $T_c$. |
| Median $T_a$ | Median of daily Ta (1984–2015) | Climatological $T_a$ for $T_c$ determination |
| $T_{diff}$ Mask | $(T_a - T_s) > 15$ K removed | Minimize cloud contaminated pixels |
| Environmental Lapse Rate (ELR) Adjustment | $(T_{a\_adj.}) = T_a - 0.003 \times (DEM - 1500)$ for DEM > 1500 m | $T_a$ reduced by 0.003 K/m above 1500 m, adjusting ELR differences between $T_s$ and $T_a$ |

NDVI = Normalized Difference Vegetation Index; $T_s$ = land surface temperature; $T_a$ = maximum air temperature; DEM = Digital Elevation Model; $T_c$ = cold/wet temperature reference limit; STD = standard deviation of $T_s/T_a$ ratio from qualifying c factor calibration pixels.

TopoWx includes the standard environmental lapse rate (ELR) for elevation in the creation of $T_a$; however, exploratory investigation found that the slope of ELR for $T_a$ was less than that of $T_s$. According to SSEBop parameterization, when $T_s$ cools faster than $T_a$, $ET_a$ will be overestimated. To solve this problem, an additional ELR correction was applied to condition the $T_a$. An elevation DEM (digital elevation model) dataset at 90-m spatial resolution was acquired from the U.S. Geological Survey (USGS)—the Global Multi-resolution Terrain Elevation Data 2010 [51]. In areas where elevation exceeded 1500 m, such as the upper and middle parts of the basin, $T_a$ was reduced by a factor of 0.003 K/m, equating to an additional reduction of 3 K for every kilometer rise. Continued analysis in lapse-rate atmospheric dynamics and conditioning for various $T_a$-to-$T_s$ relationships will aid in improving the application of SSEBop in complex, high-elevation terrain.

### 2.3. Cropland Extent

For this study, a map of the maximum cropland extent for the basin was developed to analyze water-use trends. The maximum cropland extent (484,492 ha) was derived from a combination of U.S. Department of Agriculture's (USDA) National Agricultural Statistics Services (NASS) crop data layers, National Land Cover Database (NLCD) crop zones, and maximum NDVI. The USDA-NASS provides gridded landcover/land-use classifications called crop data layers (CDL) derived from Landsat with a decision-tree classification algorithm [52]. The USDA-NASS CDL data are available CONUS-wide at 30-m spatial resolution starting in 2008 and are released annually (https://nassgeodata.gmu.edu/CropScape). To determine maximum USDA cropland extent, all CDL agricultural crop type classifications for the URGB from 2008–2015 were combined to create a maximum extent of agricultural land. Then, landcover from three NLCD products (2001, 2006, and 2011) was acquired for the basin [53–55]. The NCLD is a CONUS-wide landcover classification provided by the USGS at Landsat-scale 30-m spatial resolution. For each of the three NLCD datasets, the "Cultivated Crops" and "Hay/Pasture" zones were isolated and then combined to find the maximum cropland extent for all NCLD landcover datasets. This maximum extent was then combined with that derived from the USDA CDL to create a subsequent maximum cropland extent data layer. However, because USDA/NLCD cropland extent is based on different algorithms with their own errors and may not be uniformly accurate in all areas, it was used as a guide to filter the summer (June–August) maximum Landsat NDVI (>0.6) to find pixels actively vegetated or irrigated in each year. Yearly maximum NDVI grids were combined to find the widest extent of cropland for the entire basin. This final maximum cropland extent derived from three independent datasets was used as the sampling zone for all variables in the time-series analysis presented in this paper.

We focus here on the peak growing season of June–August when plants are actively transpiring and green. The June–August total *ETa* was estimated for the entire basin and then converted from measurements of depth in millimeter to hectare-meter (ha-m) estimates using basin area.

## 2.4. Validation of ETa Estimates using Eddy-Covariance Flux Towers

To evaluate the accuracy of the SSEBop *ETa* results, monthly modeled *ETa* estimates were compared with aggregated monthly *ETa* from eddy-covariance flux tower data provided by the AmeriFlux network (https://ameriflux.lbl.gov/) for 2007–2014. AmeriFlux is a network of Principal Investigator (PI)-managed sites measuring ecosystem $CO_2$, water, and energy fluxes in North, Central, and South America. AmeriFlux datasets are highly accepted by researchers and widely used for model calibration and validation. Six AmeriFlux sites (Figure 1c) used to evaluate SSEBop *ETa* in this study represent common ecological conditions throughout New Mexico such as desert grassland, shrubland, juniper savannah, juniper woodlands, pine forests, and subalpine conifer forests [56]. Two sites are located in high elevation forests in the Jemez Mountains of north-central New Mexico (US-Vcp: Ponderosa Pine; US-Vcm: Mixed Conifer); two lower elevation shrubland/grassland sites are located near the Sevilleta National Wildlife Refuge (US-Seg: Sevilleta grassland; US-Ses: Sevilleta shrubland); one woody savannah site is on an elevated mesa (US-Mpj: Pinyon-Juniper); and one savannah site is located in central New Mexico (US-Wjs: Juniper savannah). Based on 1980–2003 Daymet data, mean annual Daymet precipitation and temperature ranges from 244 mm of rainfall and 286 Kelvin (13 °C) at the desert sites up to 667 mm and 276 K (3 °C), respectively, at the Mixed Conifer subalpine site [56].

At each tower location, the SSEBop monthly *ETa* was sampled using a $3 \times 3$ pixel average to capture as much of the towers' footprint rather than attempting to model the exact footprint of the tower, which requires hourly wind direction information. Modeled *ETa* was compared to observed *ETa* (latent heat converted to *ETa* using heat of vaporization constant) derived from each flux tower with commonly used statistical metrics such as Pearson's r, root mean square error (RMSE), normalized root mean square error (percent derived from the range), and percent bias (from the mean) using monthly *ETa* from the flux tower and SSEBop [57]. Due to the seasonality of the data, which creates very high and low data values, the normalized RMSE was determined using the range of monthly flux tower *ETa* whereas the percent bias was determined using the mean.

## 2.5. Basin-Scale Validation of ET Estimates

To evaluate basin-scale *ETa* trends, we compared the SSEBop *ETa* estimates to the available Max Planck Institute (MPI) *ETa* during the 1986–2011 period. These monthly MPI *ETa* estimates are created using FLUXNET eddy-covariance measurements, meteorological data, climate, and the fraction absorbed photosynthetic active radiation data [58]. The MPI *ETa* estimates are coarse resolution—approximately 50-km spatial resolution—so this comparison with SSEBop *ETa* is done at the basin scale rather than the scale of cropland extent in order to understand regional trends over time. MPI provides a reasonable estimate of long-term *ETa* trends to compare to the SSEBop product and has been used to validate *ETa* trends in past studies [35,59,60]. The MPI *ETa* is only available through December 2011, so our *ETa* estimates from 2012–2015 were not compared to MPI or to basin-scale *ETa*.

We validated the SSEBop *ETa* trends at multiple spatiotemporal scales using gridded ET estimates from MPI. There are only 73 pixels from MPI that cover the entire URGB as opposed to several million with Landsat. As such, we conducted the comparison at the basin and region scale. Since the *ETa* trends presented in the study are derived from the June–August total, we also compared SSEBop totals to the MPI totals at the scale of the entire URGB as well as the basin extent within each state. We conducted Mann–Kendall (MK) trend analysis described in Section 2.6 to the MPI *ETa* to validate the same MK trends in SSEBop *ETa*.

### 2.6. Seasonality, Year-to-Year Variability, and Mann–Kendall Trend Analysis

Both water management-influenced variables (*ETa*, LST, NDVI, and runoff) and climatic variables (*ETo*, *Ta*, and precipitation) provide context for understanding ET in the URGB. Using the maximum cropland extent, the monthly variables were summarized from January 1986 to December 2015, and the 30-year mean and standard deviation for each month were calculated and presented. The goal of this analysis was to understand the variability and response of these variables to one another for each region and the entire basin. Management variables such as NDVI, LST, and runoff are explanatory variables that can help shed more light on *ETa* dynamics in relation to climatic variables such as precipitation and *ETo*.

The presence or absence of trend in the time-series data for each parameter was tested using the simple MK trend test. The MK trend test has been widely used in hydrologic data and is a non-parametric rank-based method for evaluating trends in time-series data [61–63]. The slope was measured with the Theil–Sen estimator—a non-parametric alternative to the parametric ordinary least squares regression line—which is frequently used alongside the Mann–Kendall test in hydrologic studies [64,65]. Trend analysis was performed at two scales: region/basin-wide and per-pixel level. Basin-scale analysis was carried out for the entire basin as well as each separate regional boundary—Colorado, New Mexico, Texas, and Mexico—within the URGB. Basin-scale analysis provides information on the direction and rate of change of water use. However, it does not provide information on spatial variability. Hence, we also carried out a pixel-scale analysis of trends. For each of the ~383 million Landsat pixels covering the URGB, we computed rate of change in water use and its statistical significance at 95% confidence ($p = 0.05$) level. The summaries of the pixel-based rate of change in water use are presented for croplands and the total basin area to provide context and perspective. Because the basin transects several states (and countries) and water management policies are different from region to region, for the benefit of water managers, we also present the results summarized by regions using state and country boundaries.

## 3. Results

### 3.1. Point-Scale Validation of ETa Estimates

Monthly evaluation of SSEBop *ETa* against all six sites over 2007–2014 shows a correlation coefficient "r" averaged to 0.92 and an RMSE averaged to 11.7 mm (Table 2). The strongest correlation between the SSEBop monthly *ETa* and the AmeriFlux monthly *ETa* occurs at the Juniper savannah site (r = 0.98, RMSE = 5.9 mm), but the Pinyon-Juniper (r = 0.95, RMSE = 8.8 mm) and Sevilleta shrubland (r = 0.96, RMSE = 7.4 mm) sites are also strongly correlated. The sites with the lowest correlations between SSEBop *ETa* and AmeriFlux *ETa* are the two high-elevation, forested sites Ponderosa Pine (r = 0.90, RMSE = 19.1 mm) and Mixed Conifer (r = 0.83, RMSE = 18.3 mm), but even these sites showed reasonable agreement. The comparison between SSEBop and AmeriFlux data on a monthly time-step shows strong agreement in the seasonal pattern of *ETa* (Figure 3). The average bias percentage ranges from <1% to 14% (Table 2). However, all four of the lower elevation sites show the percent bias of SSEBop in relation to AmeriFlux to be within ±3.0%.

**Table 2.** Monthly comparison of SSEBop *ETa* and Flux Tower *ETa* for New Mexico AmeriFlux Towers (2007–2014). Flux Tower and SSEBop *ETa* columns show the average of all months with matching pairs. 'r' is the Pearson's correlation coefficient and all "r" values are significant at *p* = 0.05. Normalized root mean square error (RMSE) (%) is the RMSE divided by the range of Flux Tower *ETa*. Percent Bias is the average difference between the Flux Tower and SSEBop from matching pairs divided by the average of Flux Tower *ETa*.

| Ameriflux | Site Name | Elev (m) | Flux Tower *ETa* (mm) | SSEBop *ETa* (mm) | Range (mm) | r (-) | RMSE (mm) | Normalized RMSE (%) | Bias (%) |
|---|---|---|---|---|---|---|---|---|---|
| US-Seg | Sevilleta grassland | 1596 | 27.6 | 28.1 | 89.7 | 0.91 | 10.7 | 11.9 | 1.7 |
| US-Ses | Sevilleta shrubland | 1604 | 27.1 | 26.9 | 146.7 | 0.96 | 7.4 | 5.0 | −0.7 |
| US-Wjs | Juniper savannah | 1931 | 26.4 | 27.2 | 121.9 | 0.98 | 5.9 | 4.8 | 3.1 |
| US-Mpj | Pinyon-Juniper | 2196 | 30.1 | 29.9 | 122.0 | 0.95 | 8.8 | 7.2 | −0.9 |
| US-Vcp | Ponderosa Pine | 2500 | 48.8 | 55.5 | 136.2 | 0.90 | 19.1 | 14.0 | 13.7 |
| US-Vcm | Mixed Conifer | 3030 | 40.3 | 36.0 | 108.6 | 0.83 | 18.3 | 16.9 | −10.6 |

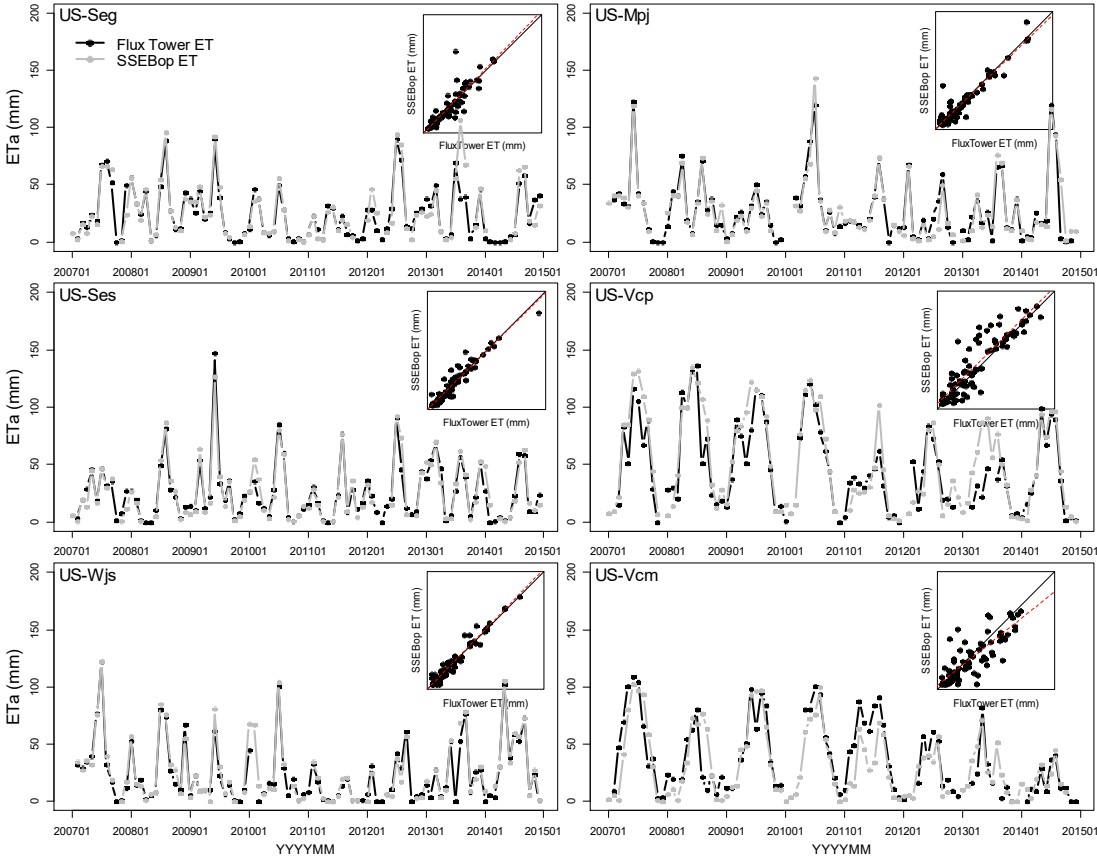

**Figure 3.** Comparison of monthly SSEBop *ETa* and observed *ETa* from the six eddy-covariance flux towers. The x-axis shows the date in 4-digit year (YYYY) and 2-digit month (MM) format.

## 3.2. Basin-Scale Validation of ETa Estimates

For basin-scale comparison against the MPI dataset, Figure 4 shows time series (1986–2011) of June–August basin average for the entire URGB for both SSEBop and MPI along with the 3-year and cumulative moving averages. Both datasets show a negative trend with a more pronounced decline during the 2002–2006 drought period. The absolute magnitudes are not directly comparable because of differences in spatial representation of the two datasets with the coarser MPI dataset sampling more areas outside of the basin boundary, which could be responsible for the lower bias in relation to the SSEBop *ETa*. Resampling the SSEBop *ETa* to match the 50-km resolution of the MPI pixels would more accurately compare the absolute magnitudes of the two *ETa* datasets, but in this context, our

interest was in looking at comparable time-series trends between the two datasets. This is a similar observation to that found in the Palo Verde Irrigated District (PVID) where the larger pixel size of MPI data included large areas of near-zero *ETa* estimates from the desert landscape outside the PVID boundary, which decreased the spatially averaged value for MPI *ETa* [35]. However, the patterns in *ETa* from 1986–2011, including the moving averages, show reasonable agreement between SSEBop and MPI *ETa* at the regional scale (Figure 4).

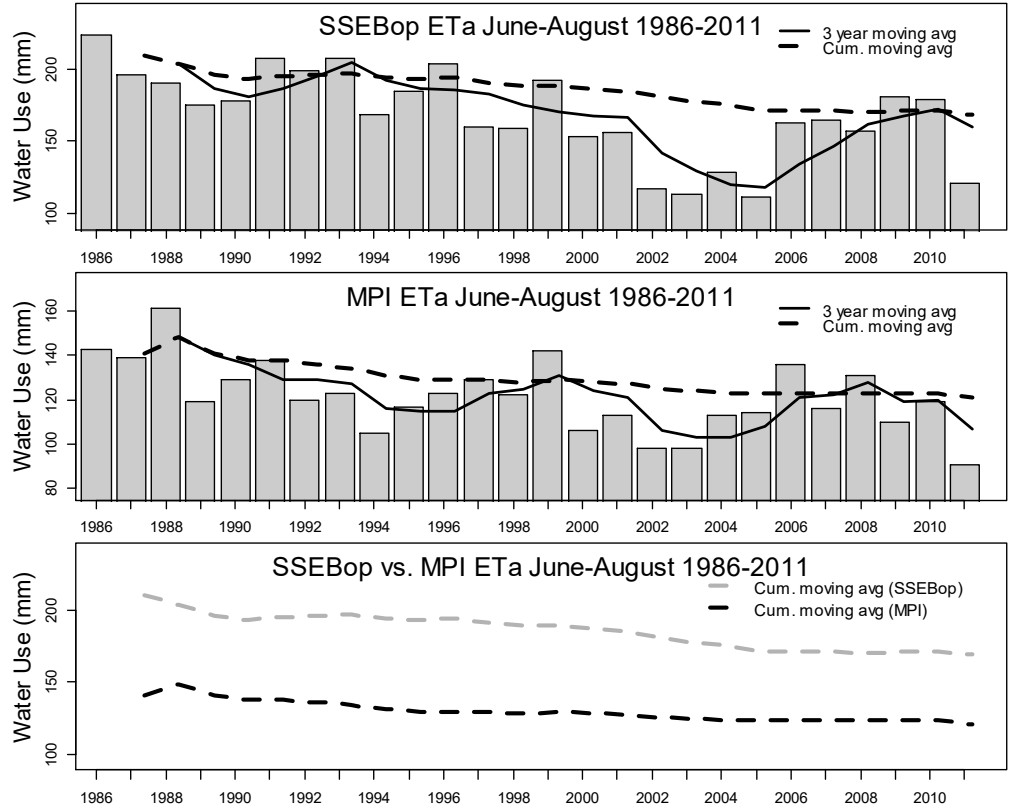

**Figure 4.** June–August *ETa* for 1986–2011 for all landcover types in the Upper Rio Grande Basin including the 3-year moving average and the cumulative average. The top chart shows the SSEBop basin averages; the middle chart shows the MPI basin averages; the bottom chart shows the cumulative moving averages for both MPI and SSEBop *ETa*.

### 3.3. Characterizing Spatiotemporal Trends in Water Use

The basin-wide URGB *ETa* shows an overall negative trend in water use for the 30-year period (Figure 5a). Average water use from *ETa* over irrigated agriculture for the 1986–2015 peak summer season is 327 mm, which converts to 158,365 ha-m of water used per year using an irrigated area (maximum extent) of 484,492 ha. Figure 5 also displays the 3-year moving average and the cumulative average over the 30-year period. As of 2015, the cumulative average for the entire basin is 158,365 ha-m. The right panel in Figure 5 indicates the deviation for each year from the 1986–2015 mean June–August *ETa*. During the 2002–2006 period, the average water use dropped to 141,850 ha-m. During the 2011–2015 period, the average water use further declined to 139,383 ha-m and was less than 136,000 ha-m in 2013, a difference of over 37,000 ha-m from the long-term median. These results once again demonstrate the capability of using remote sensing to monitor the impact of climatic and environmental stressors on water resources and landscape responses.

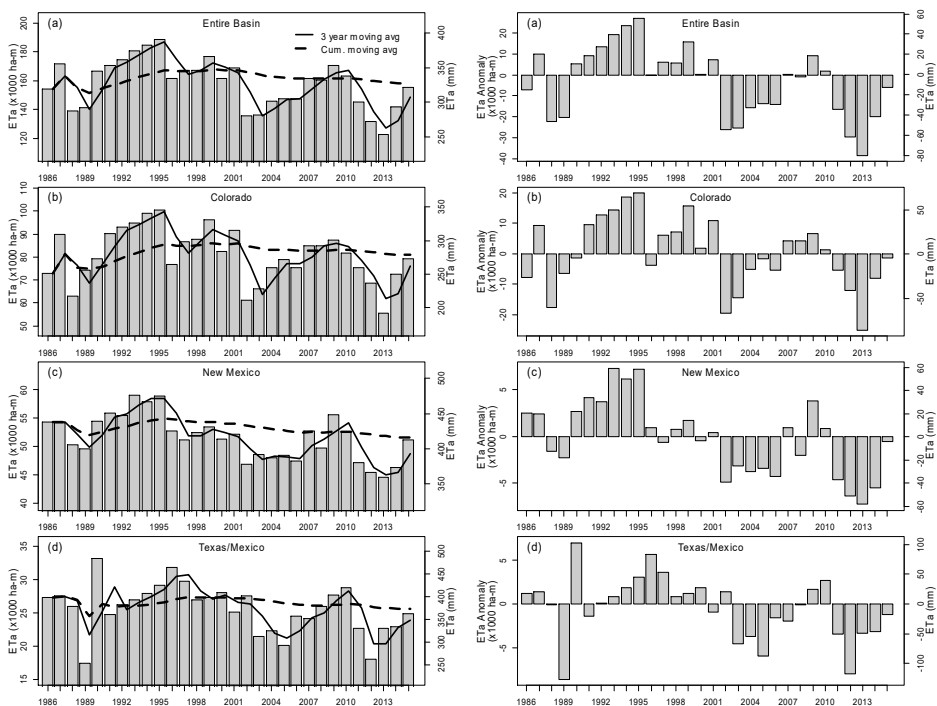

**Figure 5.** Seasonal water use and trends: (**a**) entire basin, (**b**) Colorado, (**c**) New Mexico, and (**d**) Texas/Mexico along with 3-year (solid-line) and cumulative (dash-line) moving averages and water use anomaly (right panel) by volume (ha-m) and by depth (mm).

The Colorado section of the basin has the largest percentage of the basin's total irrigated area. Over 60% of the URGB cropland extent is in Colorado with 291,469 ha, but only 51% of the total water use is used in Colorado (81,020 ha-m). Colorado shows a similar downward trend in water use over time and drops in water usage in the 2002–2006 and 2011–2015 drought periods (Figure 5b). The 3-year moving average water-use drops below 70,000 ha-m in 2004 and during 2013–2015. The average volume of water use in the 2002–2006 drought is 71,562 ha-m or a 12% reduction from the average water use of 81,020 ha-m (Table 3). The 5-year period between 2011 and 2015 saw an average volume of 70,376 ha-m of consumptive water use or 13% reduction from the average in Colorado with most of the reduction coming in 2012 and 2013, which had the sharpest reductions. The second largest reduction occurred in 2002 with 19,782 ha-m. In contrast, the 1992–1995 period shows large increases from the long-term average with an increase of 16,120 ha-m—a 20% increase. Figure 6 shows the per-pixel *ETa* anomaly for a section of Colorado as measurements of depth from the 30-year average. While the 1995 anomaly shows a general increase throughout the San Luis Valley, the 2015 anomaly shows a sharp decrease in *ETa* in the eastern section of the valley with multiple center pivot fields showing drastic reduction.

**Table 3.** Irrigated area crop water use volume and changes by region: (1) 1986–2015 average water use, (2) crop area (ha), (3) area and water use percentage of the basin, (4) the 2002–2006 average water use reduction (% and volume), and (4) the 2011–2015 average water use reduction from 30-year average (volume in ha-m and in %).

| Region | Crop Area [1] (ha) | Water Use Volume [2] (ha-m) | Percent of Basin (Area/Volume) (%) | 2002–2006 Drought Reduction ha-m/ [%] | 2011–2015 Drought Reduction ha-m/ [%] |
|---|---|---|---|---|---|
| URGB | 484,493 | 158,365 | 100/100 | −15,649 [−10] | −18,732 [−12] |
| Colorado | 291,469 | 81,020 | 60/51 | −9458 [−12] | −10,644 [−13] |
| New Mexico | 124,395 | 51,741 | 26/33 | −3730 [−7] | −4706 [−9] |
| Texas/Mexico | 68,623 | 25,651 | 14/16 | −2460 [−10] | −3383 [−13] |

[1] In hectares (ha); 1 ha = 2.47 acres. [2] In hectare meters (ha-m); 1 ha-m = 8.107 ac-ft.

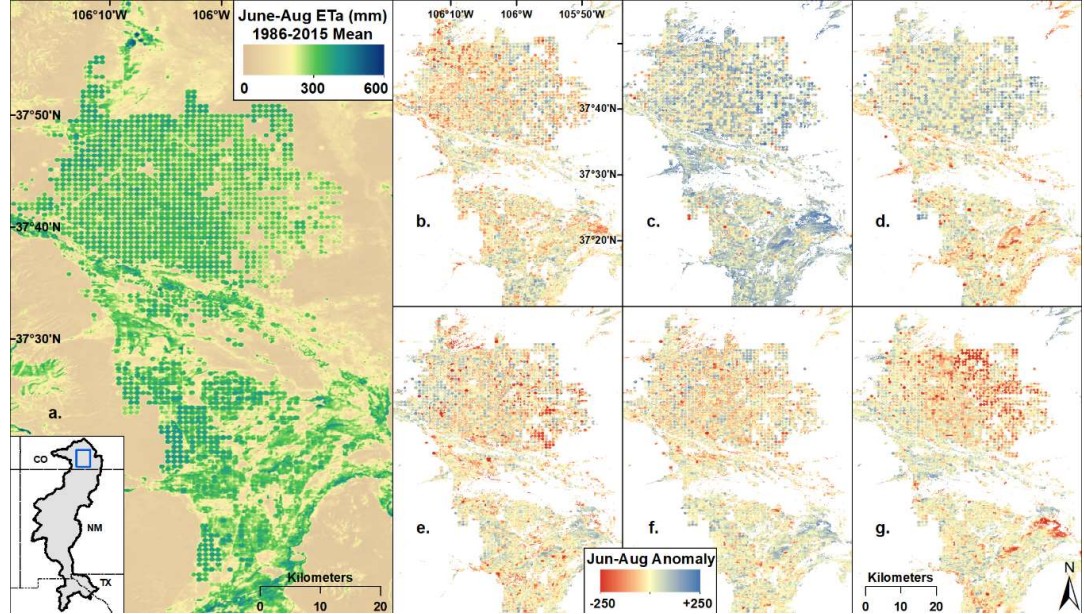

**Figure 6.** Colorado June–August *ETa* showing (**a**) 1986–2015 mean *ETa* (mm), and the annual anomaly (deviation from the mean, mm) on a 5-year interval for (**b**) 1990, (**c**) 1995, (**d**) 2000, (**e**) 2005, (**f**) 2010, and (**g**) 2015.

Although *ETa* in the New Mexico portion of the basin is not as variable as that of Colorado, there are similar patterns. The average water use for New Mexico irrigated croplands (26% of basin crop area) is 51,934 ha-m, roughly 33% of the entire basin's water use. The reductions in water use in the 2002–2006 and 2011–2015 droughts amounts to 7% (3730 ha-m) and 9% (4706 ha-m), respectively, suggesting that drought impacted the entire basin but not uniformly so, which further suggests a potential use of more groundwater in New Mexico (Table 3). The per-pixel anomaly maps in Figure 7, which displays part of the Elephant Butte Irrigation District near Las Cruces, New Mexico, shows stark decreases in many fields in 2005 and 2015, and generally with higher *ETa* (positive deviation) in 1990, but showing relatively more stable *ETa* values as compared to Figure 6. Although there is less variation in year-to-year *ETa* volume in New Mexico than in Colorado, there is still a slight downward trend in *ETa* over time (Figure 5c).

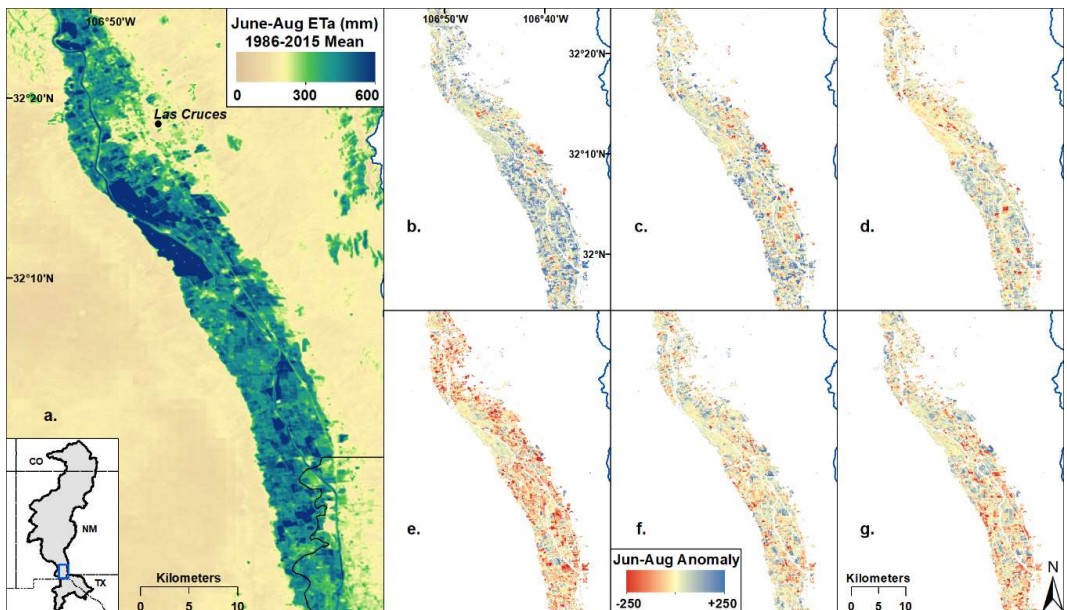

**Figure 7.** New Mexico June–August *ETa* showing (**a**) 1986–2015 mean *ETa* (mm), and the annual anomaly (deviation from the mean, mm) on a 5-year interval for (**b**) 1990, (**c**) 1995, (**d**) 2000, (**e**) 2005, (**f**) 2010, and (**g**) 2015.

The remaining irrigated areas in the URGB are located south of El Paso, Texas, and straddle the border between Texas and Juarez, Mexico. As shown in Figure 8, on the US side of the border, the *ETa* is close to the 30-year average or above it for many fields in 1990, 1995, and 2000, but drastic reductions from the average (higher anomalies) occur in 2005 and 2015, especially, during the drought periods. In the 1990 and 1995 anomaly maps, there are clear differences between the Mexico side of the border and the US side in terms of anomaly from the 30-year mean. This difference may be due to varying water management decisions and water resource allocation; however, after 2000, the *ETa* signal on both sides of the border becomes more consistent, suggesting similar water management decisions.

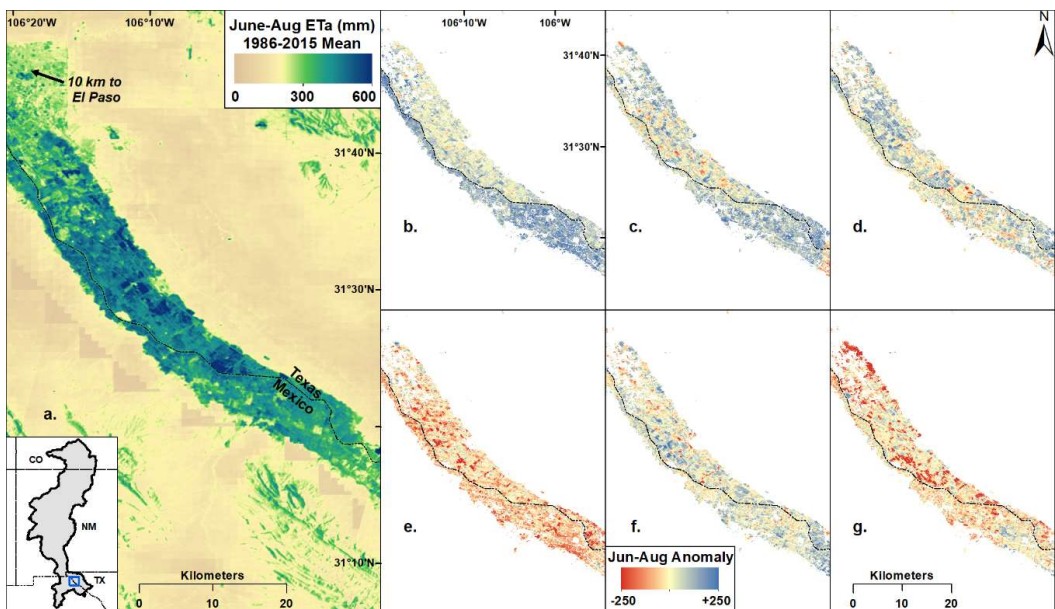

**Figure 8.** Texas and Mexico June–August *ETa* showing (**a**) 1986–2015 mean *ETa* (mm), and the annual anomaly (deviation from the mean, mm) on a 5-year interval for (**b**) 1990, (**c**) 1995, (**d**) 2000, (**e**) 2005, (**f**) 2010, and (**g**) 2015.

The irrigated areas in Texas and Mexico are only 14% of the total irrigated area in the URGB with a combined area of 68,623 ha. However, the Texas and Mexico irrigated area accounts for roughly 16% of the water use in the basin with median water use of 25,651 ha-m (Table 3). Compared to the upper basin areas, the lower basin *ETa* shows higher year-to-year variability but still displays a general downward trend over time including reduction in water use during the early 2000s and the 2011–2015 period, suggesting that drought stressors, unsurprisingly, continue to be felt downstream (see Figure 8).

### 3.4. Characterizing Seasonality of Agro-Climatic/Hydrologic Variables within Agricultural Lands

Each of the seven variables—both water management-influenced and climatic variables in the Colorado irrigated areas—show distinct seasonal patterns of low activity in the winter months and a high peak in the summer months (Figure 9). Maximum air temperature and *ETo* show strong seasonal patterns with characteristic summer peaks. Annual precipitation is very low with a peak in mid-summer. NDVI remains low for much of the year except between the months of June through August. LST warms rapidly in the spring, then declines in June and July—this decline in the summer can be attributed to the evaporative cooling of vegetation, a pattern not shared by maximum air temperature. This decline is a testament to the fact that LST is impacted by management practices—in this case, irrigation. The seasonal pattern of SSEBop *ETa* in the SLV is very low in magnitude but shows a characteristic seasonal pattern of very low *ETa*, down to almost background (precipitation levels) in the winter months before rising in May and peaking in July.

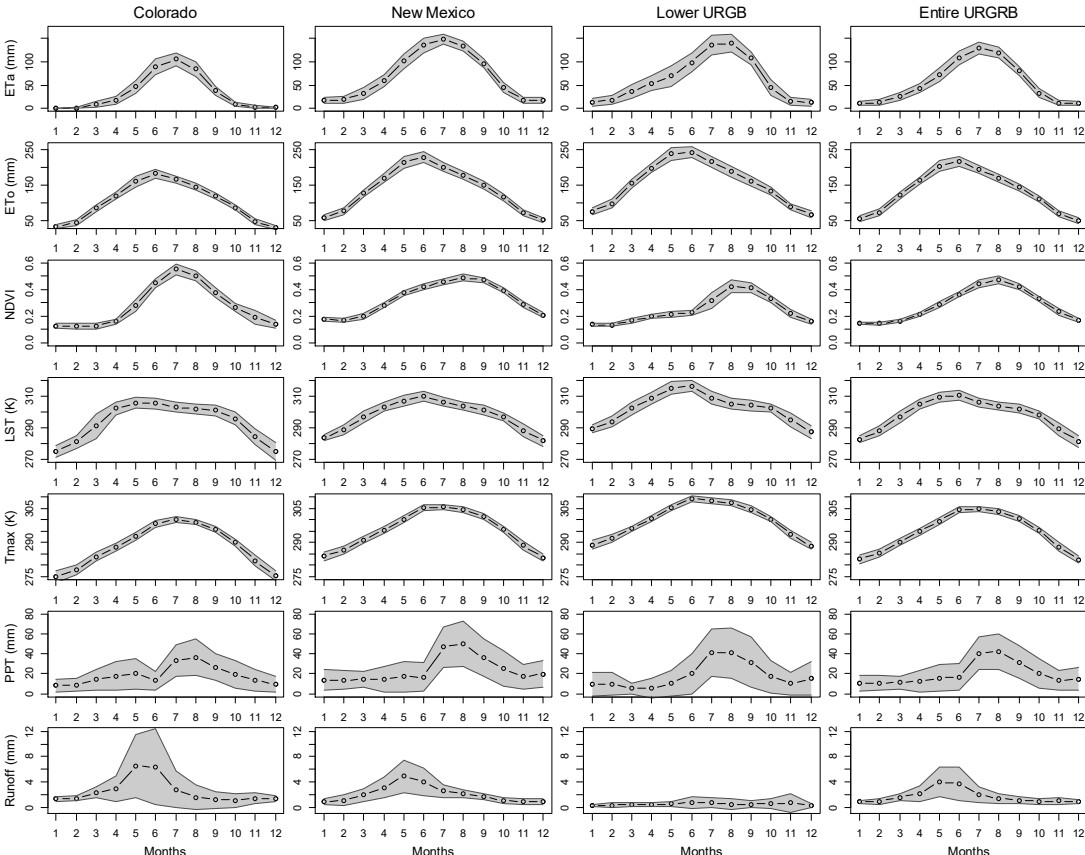

**Figure 9.** Seasonal patterns of seven agro-hydrologic variables for the cropland portions within each region boundary. The 1986–2015 monthly mean with one standard deviation (in dark grey) for actual evapotranspiration (*ETa*), reference ET (*ETo*), normalized difference vegetation index (NDVI), land surface temperature (LST), maximum air temperature (*Ta*), precipitation (PPT), and runoff.

Whereas the seasonal patterns of climatic variables in New Mexico are similar to Colorado, the management-influenced variables in New Mexico are noticeably different. NDVI suggests a slightly longer growing season in New Mexico as opposed to the SLV in Colorado, with irrigated cropland in New Mexico showing higher NDVI from May through September. Conversely, LST for irrigated cropland in New Mexico rises sharply but peaks in June when crops are developing and evaporative cooling brings the LST back down again through the summer months. The SSEBop *ETa* has a characteristic seasonal pattern, rising in spring to above 100 mm/month in May with more variability in the early summer until peaking in July (nearly 150 mm/month) and finally falling below 100 mm/month in September, suggesting a May–September prime growing season.

In the lower basin regions (Texas-Mexico), *ETo* and *Ta* both have high magnitudes, most likely due to the lower latitude and elevation, but the overall seasonal pattern does not radically change (Figure 9). As shown in Figure 9, *ETa* tends to peak between *ETo* and NDVI, confirming the dependence of *ETa* on weather (*ETo*) and vegetation/soil moisture conditions, which is comparable to the Idaho-based study previously reported [66].

### 3.5. Mann–Kendall (MK) Trend Analysis

### 3.5.1. Basin-Scale Mann–Kendall Trend Analysis

The basin-scale MK analysis was performed on each of the seven hydro-climatological variables. Results from the MK analysis conducted on the seasonal time-series data for the entire basin and regions are presented in Table 4. At the entire basin scale, the climatic parameter *Ta* showed a significant positive trend, but *ETo* did not show a statistically significant trend. Precipitation (PPT) showed a statistically significant negative trend at $p = 0.1$ level for the Colorado region and the basin, but PPT showed no statistically significant trend for the lower drier regions. SSEBop *ETa* showed a statistically significant negative trend at 95% confidence for the basin overall as well as in New Mexico and Texas, but not in Colorado. NDVI showed no statistically significant trend, but the cautionary notes from Roy et al. [67] and Ke et al. [68] on the inconsistency of top-of-atmosphere NDVI from different sensors in the Landsat record, such as the higher bias in Landsat 8 NDVI, may need to be examined, which is beyond the scope of this study.

The negative MK trends for SSEBop *ETa* are corroborated by a comparison with the MK results for the MPI datasets. The MK test was conducted on the MPI *ETa* from 1986–2011 on the basin and region extent for all cropland and non-cropland pixels as the MPI does not have sufficient spatial resolution to differentiate crops from other landcover. The same MK test was run for the SSEBop *ETa* at the same spatiotemporal scale and the results are presented in Table 5 (also see Figure 4 for the time series comparison for the basin). The MK results for the entire URGB *ETa* demonstrates a statistically significant negative trend for both the MPI *ETa* and the SSEBop *ETa* for all pixels. At the region scale, the MK results for both MPI and SSEBop generally agree and show a statistically significant negative trend for both New Mexico and Texas. For the basin extent in Colorado, there is a statistically significant negative trend in MPI, but although the slope is negative for SSEBop *ETa*, it is not statistically significant. Similarly, both MPI and SSEBop show a statistically negative trend in the Mexico section of the basin at $p = 0.1$.

### 3.5.2. Pixel-Scale Mann–Kendall Trend and Rate of Change Analysis

Pixel-scale analysis provides field-level information on the spatial variability of crop water use trends across the URGB. Figure 10 shows the 30-year average SSEBop *ETa* along with the slope grid in Figure 10b and the slope filtered by statistically significant pixels as rate-of-change in mm per decade. Unsurprisingly, most of the non-irrigated cropland pixels show little or no statistically significant trend in the MK test. The areas of significant increase (>100 mm/decade) in seasonal water use appear to be the center pivots, which were established during the study period (1986–2015), or a switch to crops with relatively higher water demand. Similarly, some areas/center pivots of significant decline (<−100

mm/decade) could be areas of abandoned agriculture or crop type changes that lead to a negative trend in crop water use. There are also many areas on the boundaries of and in between center pivots that show statistically significant decreases in crop water use, which could suggest improving irrigation efficiency, but this would need to be confirmed with more data.

**Table 4.** Mann–Kendall trend analysis (June–August 1986–2015) of the spatially averaged agro-climatic/hydrologic parameters summarized for entire basin and regions.

| Basin/State | | Climate Parameters | | | Management Parameters | | | |
|---|---|---|---|---|---|---|---|---|
| | | $ETo$ (mm) | $Ta$ (K) | PPT (mm) | $ETa$ (mm) | LST (K) | NDVI (-) | $Q$ (mm) |
| Entire Basin | $p$ | 0.153 | 0.000 | 0.087 | 0.030 | 0.001 | 0.775 | 0.225 |
| | Slope [1] | 7.0 | 0.6 | −9.5 | −13.9 | 1.8 | 0.00 | −1.3 |
| | Trend [2] | (#) | (+) ** | (−) * | (−) ** | (+) ** | (#) | (#) |
| Colorado | $p$ | 0.695 | 0.001 | 0.064 | 0.125 | 0.001 | 0.943 | 0.412 |
| | Slope [1] | 2.6 | 0.7 | −12.1 | −13.9 | 1.7 | 0.00 | −1.3 |
| | Trend [2] | (#) | (+) ** | (−) * | (#) | (+) ** | (#) | (#) |
| New Mexico | $p$ | 0.050 | 0.001 | 0.269 | 0.001 | 0.000 | 0.568 | 0.035 |
| | Slope [1] | 14.1 | 0.6 | −9.8 | −23.4 | 1.8 | 0.00 | −1.6 |
| | Trend [2] | (+) ** | (+) ** | (#) | (−) ** | (+) ** | (#) | (−) ** |
| Texas | $p$ | 0.017 | 0.001 | 0.775 | 0.035 | 0.000 | 0.009 | 0.019 |
| | Slope [1] | 18.1 | 0.5 | −3.9 | −33.4 | 3.1 | −0.03 | −0.4 |
| | Trend [2] | (+) ** | (+) ** | (#) | (−) ** | (+) ** | (−) ** | (−) ** |
| Mexico | $p$ | 0.035 | 0.001 | 0.643 | 0.544 | 0.029 | 0.830 | 0.038 |
| | Slope [1] | 16.9 | 0.5 | −5.1 | −10.5 | 1.7 | 0.00 | −0.3 |
| | Trend [2] | (+) ** | (+) ** | (#) | (#) | (+) ** | (#) | (−) ** |

[1] Seasonal rate of change in mm/decade. [2] (+) indicates significant positive trend; (−) indicates significant negative trend; * indicates statistical significance at $p = 0.1$ and ** indicates significance at $p = 0.05$; (#) indicates that there is no significant trend.

**Table 5.** Mann–Kendall trend analysis (June-August 1986–2011) of the spatially averaged $ETa$ from SSEBop and from MPI for the entire basin and regions for all cropland and non-cropland pixels.

| Basin/State | | SSEBop $ETa$ (1986–2011) | MPI $ETa$ (1986–2011) |
|---|---|---|---|
| Entire Basin | $p$ | 0.001 | 0.008 |
| | Slope [1] | −27.0 | −11.0 |
| | Trend [2] | (−) ** | (−) ** |
| Colorado | $p$ | 0.103 | 0.004 |
| | Slope [1] | −9.0 | −5.7 |
| | Trend [2] | () | (−) ** |
| New Mexico | $p$ | 0.001 | 0.025 |
| | Slope [1] | −33.0 | −12.0 |
| | Trend [2] | (−) ** | (−) ** |
| Texas | $p$ | 0.015 | 0.047 |
| | Slope [1] | −30.0 | −15.0 |
| | Trend [2] | (−) ** | (−) ** |
| Mexico | $p$ | 0.022 | 0.058 |
| | Slope [1] | −31.0 | −13.0 |
| | Trend [2] | (−) ** | (−) * |

[1] Seasonal rate of change in mm/decade. [2] (+) indicates significant positive trend; (−) indicates significant negative trend; * indicates statistical significance at $p = 0.1$ and ** indicates significance at $p = 0.05$; (#) indicates that there is no significant trend.

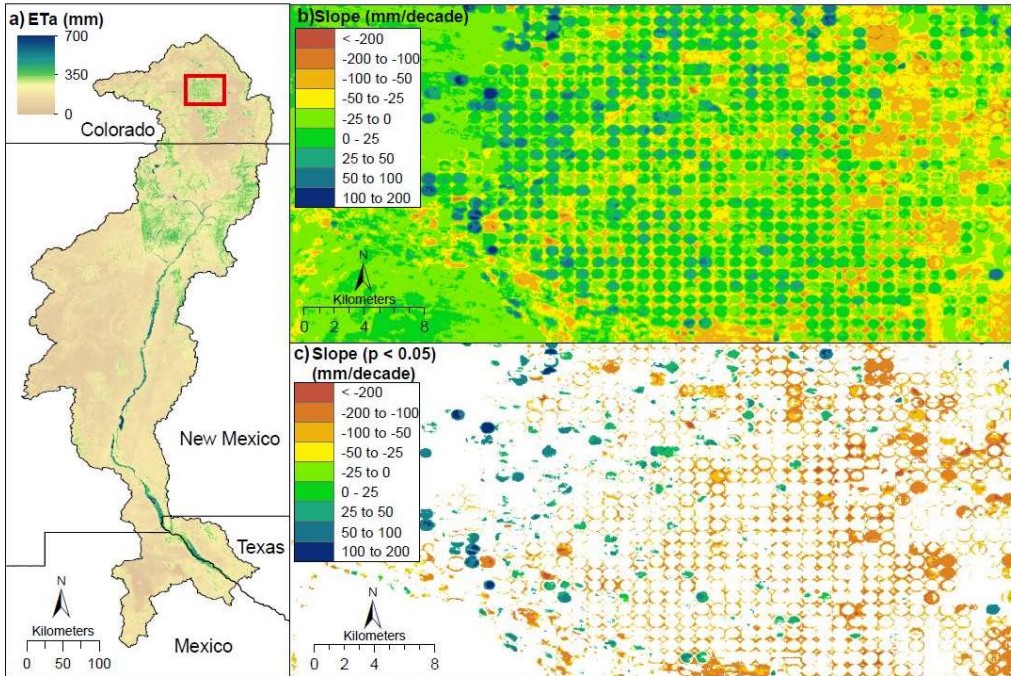

**Figure 10.** Pixel-based rate of change in seasonal crop water use (*ETa*) for center pivots in the Colorado part of the URGB. (**a**) Landsat based summer median (1986–2015) *ETa*, (**b**) rate of change (mm/decade) in seasonal (June–August) crop water use (*ETa*), and (**c**) rate of change (mm/decade) of ET for only statistically significant pixels using pixel-based MK analysis at $p < 0.05$.

Figure 11 displays the histograms for areas with statistically significant slope ($p < 0.05$) for the irrigated cropland pixels but also for all pixels of all landcover types, which was included for perspective. The results indicate that 17.8% of all landcover type pixels in the basin showed negative change, and <1% of the basin area showed significant (positive) increase in seasonal water use (Figure 11, top left panel) with the remaining pixels showing no statistically significant trend. The smaller regions showed similar patterns for all landcover types with less than 1% showing positive trends and between 10–20% negative trends, depending on region (see Figure 11, left column).

When filtering for only irrigated cropland pixels, there is a similar pattern with most of the statistically significant cropland pixels showing a negative trend as opposed to a positive trend (Figure 11, second column). The histogram of the cropland pixels for the URGB showed that seasonal water use of about 30% of the total cropland area as decreasing, about 7% of the croplands showed a positive trend, and remaining pixels showed no significant trend or change in water use. The croplands in the Colorado part of the URGB showed over 26% negative trends as opposed to 7% with positive trends. However, the lower regions of the basin showed more drastic change with nearly 38% of cropland pixels in New Mexico showing a negative trend as opposed to less than 6% showing a positive trend. Similarly, Texas and Mexico showed negative trends for 45% and 32% of the cropland pixels, respectively, with only slightly more than 3% of cropland pixels in Texas showing a positive trend over time. Overall, the Texas cropland showed the most relative water use decline by area (# pixels) in relation to the increase when compared to other regions in the basin. This could be the combined effect of reduced water availability from ground and surface water resources.

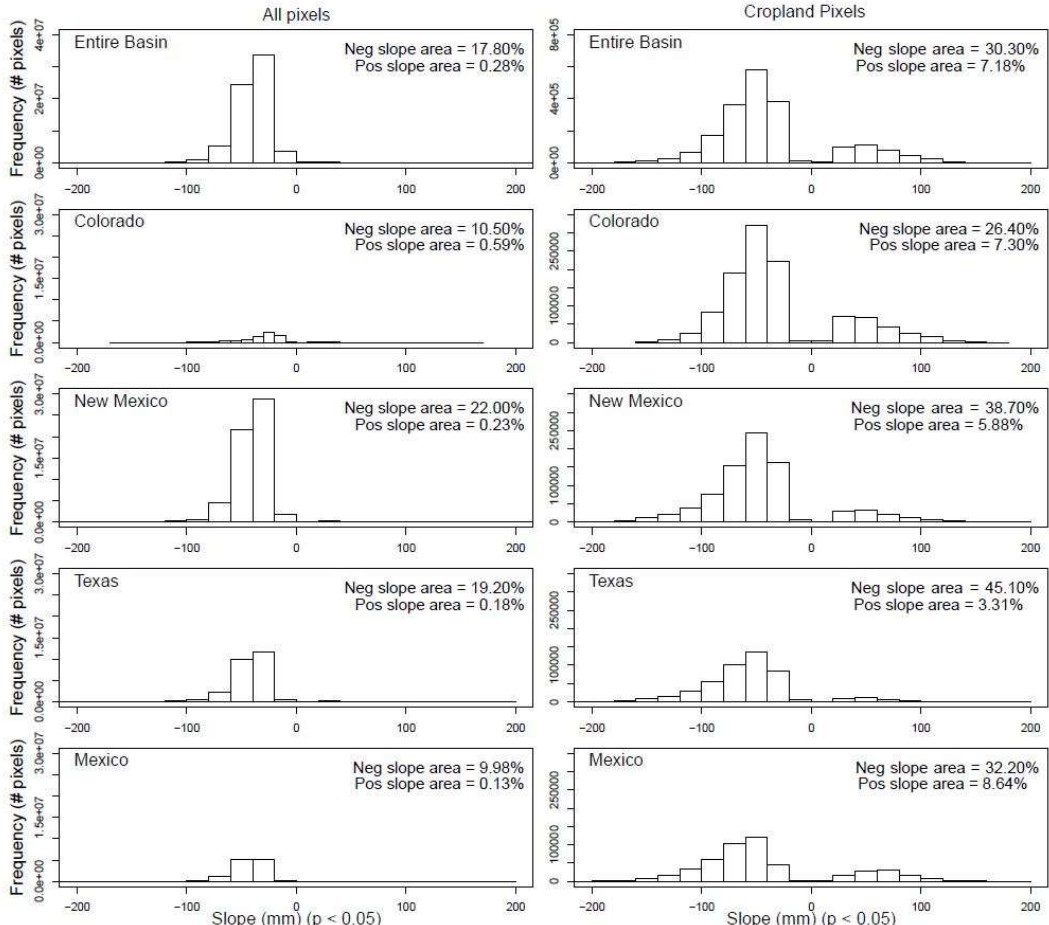

**Figure 11.** Histogram plots showing frequency distribution of pixels with significant negative (neg) or positive (pos) slope ($p < 0.05$) for *ETa* over the entire basin or regions. Left panel shows all pixels (all land cover types) and right panel is only for cropland areas. Note that the percentages do not add up to 100% as the neutral (non-significant changes) areas are not shown.

## 4. Discussion

The increasing promise of remotely sensed data for updating existing records and routinely monitoring agricultural water use is a priority for water managers and planners around the world. The USGS Water Use and Availability Program (https://water.usgs.gov/watercensus/index.html) is currently exploring the use of remote sensing to compile and monitor water use components in the United States at the level of local watersheds using the Hydrologic Unit Code (HUC-12) hydrologic divisions determined by the Watershed Boundary Dataset (WBD) (https://datagateway.nrcs.usda.gov).

The usefulness of remote sensing data is more apparent in capturing spatial dynamics and seasonal monitoring. The challenge has been in the use of remote sensing data for historical studies and trend analysis due to the relatively short history (1984–present) and changes in sensors over time. Consistent methods that are applicable to multiple sensor types are vital for accurate historical trend analysis.

Although *ETa* trend studies at regional scales have been conducted using AVHRR (since 1970s) and MODIS (since 2000) sensors, no historical field-scale study has been reported using Landsat-based *ETa* for crop water use studies [24,25]. Existing long-term crop water use studies have focused on the use of crop coefficients (Kc) derived from NDVI for specific crops, which assumes optimum agricultural practices [27–29]. And as Roy et al. [67] point out, NDVI may vary by sensor in the Landsat record and should not be assumed to be consistent over time. As Samani et al. [31] reported, even irrigated agriculture is not 100% efficient in water use for various reasons ranging from poor understanding of irrigation scheduling to availability. Thus, the use of the direct observation of *ETa* with minimal

assumptions is critical for capturing spatiotemporal variability of water use under all (optimal or non-optimal) agricultural practices.

The use of the SSEBop model for this historical study brings an advantage over use of crop coefficient methods for the following reasons: (1) There is no assumption on crop types and thus eliminates uncertainty associated with crop-type classification; and (2) there is no assumption on optimum crop management, which could exaggerate crop water use. However, the SSEBop model is still prone to uncertainties from input data quality, especially from cloud contamination and unequal number of images over different years. Nevertheless, SSEBop is a robust *ETa* model as it relies on two model parameters ($\gamma^s$ and *Tc*) that rely on average climatic data and handles sensor differences through the c factor parameter. Potential differences in LST among Landsat 5, 7, and 8 are minimized using the scene-based c factor, which helps evaluate every LST image for corresponding environmental conditions between the wet (*Tc*) and dry limits (*Tc* + $1/\gamma^s$). Furthermore, validation results with six eddy-covariance sites and with the independent MPI gridded flux datasets support the reliability of the SSEBop estimates. While the flux tower comparison provides a suite of accuracy metrics, the MPI supports the overall negative *ETa* trend in the URGB.

This study presents trends and volumetric magnitudes on the different parts of the basin using state boundaries as one way of summarizing regional variability. It is important to note that the volumetric water use estimates are only presented for the 3-month peak-summer period and thus should not be used as annual magnitudes. The fact that the order of volumetric water use percentages of 51%, 33%, and 16% correspond with Colorado, New Mexico, and Texas/Mexico region of the basin, respectively, is not surprising as it follows the order of crop area percentages with the corresponding 60%, 26%, and 14%, respectively. However, it is interesting to note that in terms of water use per unit area, the Colorado region uses the least and New Mexico uses the most. This spatial difference in water use is related to the regional climate, with the cooler upper basin using only 51% of the water while occupying 60% of the crop area. New Mexico and Texas/Mexico are more comparable, both having a higher percentage of water use than their crop area proportion with New Mexico having slightly more than Texas/Mexico.

Although both SSEBop (1986–2015) and MPI (1985–2011) show an overall negative trend (more persistent since 1995), the 3-year moving average shows a more cyclic water use pattern, most likely affected by drought events that could influence water availability. In regional-scale analysis, only New Mexico and Texas showed a statistically significant negative trend, which is supported by the decline in runoff in the same region. This trend suggests declining surface water is leading to a decline in irrigation-based *ETa*. On the other hand, the Colorado region did not show a statistically negative trend in *ETa* in the upper basin, potentially corroborated by a neutral trend in runoff (Table 4).

As expected, during drought periods, the *ETa* from all regions showed a reduction compared to the average year, but the rates of decline suggest a differential impact of drought in different regions of the basin. The water use decline ranged from 7% in New Mexico (2002–2006) to 13% in Colorado, Texas/Mexico (2011–2015); the New Mexico region showed the least decline compared to the other regions on the two drought periods. Understanding the causes for the variability in drought impacts requires more information on the water management strategies, including conjunctive use of groundwater resources in the basin and potential shifts in crop types over time and space.

The pixel-based trend analysis reveals two pieces of information: (1) Most of the pixels did not show significant trends both in basin-wide (82%) and cropland analysis (63%); and (2) pixels show most negative trends at the basin scale, but this varies in crop areas (Figure 11). The neutral trend observed in all landcover pixels of the basin could be explained by the vast desert area in the basin, which shows little change in *ETa*, and/or the changes are too small to be captured by the resolution of the data and the accuracy of the model. The statistically neutral cropland pixels are most likely tied to the consistency of irrigation over those areas across the years. The mix of positive trend (7%) and negative trend (30%) pixels over cropland areas could be attributed to the expansion and abandoning of irrigated areas or change in crop types with varying *ETa* rates. The percent of neutral pixels over

crop areas appears to generally decrease as we move south with 66%, 55%, 52%, and 59% associated with Colorado, New Mexico, Texas, and Mexico regions, respectively, suggesting stability of irrigation practices with increasing proportion of neutral pixels.

Mann–Kendall trend analysis of the seven agro-climatic/hydrologic parameters was useful in partially explaining the trends observed in *ETa*. It is interesting to note that the MK tests for NDVI shows no statistically significant trends, except for the irrigated area in Texas. However, the Theil–Sen's slope of NDVI is slightly negative, which is in line with the increasing slope of LST, as these two variables show strong inverse correlation in peak (June–August) summer with statistically significant r of −0.54 and −0.73 for Colorado and New Mexico, respectively (not shown). When looking at year-to-year change in Colorado, the average change in NDVI approaches zero whereas the average change in LST is +0.14 K. However, when looking at a more drastic change in a drought year, such as 2002, we see NDVI drops 0.12 from 2001, while LST increases from 301 K in 2001 to over 309 K in 2002, an increase of over 8 K. Even in a drought year where drastic change is expected, the NDVI does not show as much change as LST, which demonstrates that LST is more dynamic than NDVI.

On the other hand, the negative trend in runoff for the lower basin areas may be attributed to changes in precipitation, which showed a statistically negative trend at $p = 0.1$ level in the relatively abundant precipitation upstream region of Colorado and the basin at large. The decreasing runoff can explain the reduction in *ETa* in these lower regions. However, the conjunctive use of surface and groundwater for irrigation complicates the effort to connect the decreasing runoff to changes in irrigation practices.

## 5. Conclusions

The main objective of this study was to estimate, map, and analyze historical (1986–2015) crop water use dynamics in the Upper Rio Grande Basin (URGB) over a 30-year period. The study used peak-summer (June–August) *ETa* to characterize and assess water use trends in different parts of the URGB.

Review of existing literature indicated the lack of research on the application of historical Landsat thermal data for long-term crop water use studies and trend analysis. The comparison of SSEBop *ETa* to *ETa* derived from six eddy-covariance flux towers, located in diverse agro-climatic regions, over an 8-year period (2007–2014) showed a strong correlation between the remote-sensing *ETa* and land-based observed *ETa*. SSEBop *ETa* showed an average "r" of 0.92 and RMSE of 11.7 mm. The average percent bias in monthly *ETa* between SSEBop and AmeriFlux ranges from 14% at the most extreme to less than 1%. Furthermore, the gridded FLUXNET data from MPI corroborated the SSEBop *ETa* trends by showing significant negative trends at basin and regional scales. These independent evaluations demonstrate the reliability of SSEBop *ETa*, derived from Landsat imagery, for conducting a spatiotemporal characterization of *ETa* in diverse agro-climatic settings.

Overall, the seasonal (June–August) crop water use in the URGB showed a decline over the 30 years on different parts of the basin: for the entire basin, the seasonal water use declined at a rate of −13.9 mm/decade, which is equivalent to a 12.8% percent reduction from the 30-year average water use (−41.7 mm out of a basin-wide 327 mm/season). The Texas region showed the highest decline at a rate of −33.4 mm/decade followed by New Mexico (−23.4 mm/decade) while the Colorado and Mexico regions did not show a significant trend. Furthermore, per-pixel trend analysis revealed much of the observed trend in the basin is attributable to less than 20% (all land cover type) or less than 40% (cropland) of the area while the rest of the basin remained neutral, suggesting stability of irrigation practice over the majority of irrigated fields.

The volumetric seasonal water use has shown a consistent reduction during 2002–2006 and 2011–2015 that included known drought years, suggesting the decrease in water use may be linked to surface water availability as it is supported by a negative trend in runoff. However, drought seems to have differential impact across the basin, with the least decrease (7%) observed in New Mexico and the highest decrease (13%) in Colorado, as well as the Texas/Mexico regions during the two drought

periods. Further investigation is planned to determine the relative contributions of management and climate drivers in the observed negative water-use trend using more detailed water budget studies and analysis of changes in cropping patterns in space and time.

This study has demonstrated the consistency and usefulness of historical Landsat data (Landsat 5, 7, and 8) and the robustness of the SSEBop *ETa* model for understanding the spatiotemporal dynamics of water use in a relatively large and complex basin. This approach is scalable to any basin in the world using Landsat and associated climatic datasets owing to advances in cloud-computing resources, which can allow nationwide and global applications of SSEBop *ETa*. Irrigation managers, water resource planners, and policy makers can make best use of these powerful datasets and tools for better management of scarce water resources.

**Supplementary Materials:** Monthly SSEBop ETa from 1984–2015 for the Upper Rio Grande Basin is publicly available from the USGS at: https://earlywarning.usgs.gov/ssebop/landsat/605.

**Author Contributions:** Conceptualization: G.B.S. and M.S.; methodology: G.B.S., M.S., and M.F.; software: M.S. M.F., and S.K.; validation: R.K.S., M.S., N.M.V., and M.L.; formal analysis: G.B.S., M.S., and N.M.V.; investigation: G.B.S., M.S., and N.M.V.; data curation: M.S., N.M.V., R.K.S., S.K., M.F., M.L., and K.R.D.-M.; writing—original draft preparation: G.B.S., M.S., and N.M.V.; writing—review and editing: G.B.S., M.S., N.M.V., R.K.S., S.K., M.F., M.L., and K.R.D.-M.; visualization: M.S., N.M.V., and S.K.; supervision: G.B.S.; project administration: G.B.S.; funding acquisition: G.B.S.

**Funding:** This work was performed under U.S. Geological Survey (USGS) contracts G15PC00012 and #140G0119C0001 in support of the USGS Water Census program though the USGS Land Change Science funding mechanism. Flux tower funding was provided through DOE Ameriflux Mangement Project Subcontract No. 7074, National Science Foundation Grant EAR-1331408 in support of the Catalina-Jemez Critical Zone and NSF-DEB award to the University of New Mexico for Long-Term Ecological Research (Sevilleta LTER).

**Acknowledgments:** This study is part of the U.S. Department of the Interior's WaterSMART (Sustain and Manage America's Resources for Tomorrow) and USGS Land Chance Science programs. All data used in this study were obtained from public domains and are freely available. We thank the three anonymous journal reviewers and Matthew Rigge of EROS for their insightful comments and suggestions. We are grateful to Sandy Cooper of USGS for her excellent edits and editorial comments and suggestions. Any use of trade, firm, or product names is for descriptive purposes only and does not imply endorsement by the U.S. Government. All the data generated in this study are available at: https://www.sciencebase.gov/catalog/item/5bd86bc2e4b0b3fc5ce9f87e (accessed July 03, 2019).

**Conflicts of Interest:** The authors declare no conflict of interest. The funders had no role in the design of the study; in the collection, analyses, or interpretation of data; in the writing of the manuscript, or in the decision to publish the results.

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
