# Peer review of "Long-Term (1986–2015) Crop Water Use Characterization over the Upper Rio Grande Basin of United States and Mexico Using Landsat-Based Evapotranspiration"

_remotesensing, doi:10.3390/rs11131587_

Round 1

Reviewer 1 Report

1.     The scaling coefficient k is set as 1.25 in this study. However, k is set as 1.2 in other study by the author. Please give an explanation.

2.     Line 255-257. It seems the ETa is linear interpolated for cloud-masked pixels. As I known, evaporative fraction or reference evaporative fraction is linear interpolated for cloudy pixels and then ET is calucated with Rn or ETo in METRIC/SEBS. So I guess ETf should be interpolated first.

3.     Land surface temperature is a key parameter for SSEBop. I wondered how large uncertainty would cause by the missing value in remote sensing ?  For point-scale validation, I would suggest to add how many LST values are available.

4.     The MPI ETa datasets was adopted to comparison against with SSEBop ET estimates. However, the spatial resolution of the MPI ETa datasets is too coarse. I would suggest to use GLDAS products as a replacement.

5.     The mass balance approach should be used in Basin-scale validation of ET.

6. How the missing values in remote sensing would have effects on the trend analysis?

Author Response

Reviewer 1:

1.     The scaling coefficient k is set as 1.25 in this study. However, k is set as 1.2 in other study by the author. Please give an explanation.

Author’s response:

This is a correct observation.  The main reason for this difference is changes in model parameterization that brings compensatory multiplying factors.  SSEBop has two model parameters dT (differential temperature) and Tc (wet/cold reference limit). We have been improving on the determination of these parameters. The changes in these parameters have an effect on the “k” factor. Besides, our comparison with flux tower suggests k = 1.25 is an appropriate scaler.  Other sources of differences is the source of the ETo. In this study, we used GRIDMET ETo which could be different from other sources of ETo we used in the past. Such changes in data type and model parameter are expected to bring bias which could be adjusted through the k factor.

 2.     Line 255-257. It seems the ETa is linear interpolated for cloud-masked pixels. As I known, evaporative fraction or reference evaporative fraction is linear interpolated for cloudy pixels and then ET is calucated with Rn or ETo in METRIC/SEBS. So I guess ETf should be interpolated first.

Author’s response:

That is correct! And that is what we did in this study where ETf was interpolated first. We have clarified this in the revised manuscript with wording to make clear that we interpolated ETf before computing the total ETa

3.     Land surface temperature is a key parameter for SSEBop. I wondered how large uncertainty would cause by the missing value in remote sensing ?  For point-scale validation, I would suggest to add how many LST values are available.

Author’s response:

It is true LST is the most improver forcing variable in SSEBop. Thus, number of clear (non-cloudy) images determines the accuracy of monthly total estimates. Prior studies have reported that seasonal ET can be reasonably estimated with one clear image per month.  For our point- scale validation exercise, we only included those months with at least one clear image per month.  But we have not investigated the incremental improvements from having more images per month in this study, which is an important and interesting study by itself.

 4.     The MPI ETa datasets was adopted to comparison against with SSEBop ET estimates. However, the spatial resolution of the MPI ETa datasets is too coarse. I would suggest to use GLDAS products as a replacement.

Authors response: MPI ETa has been produced by empirically upscaling global FLUXNET ETa and machine learning approaches and has been validated around the world [1]. MPI ETa has been thoroughly validated against other ET products and was found to be a reliable source of ETa globally [2,3].

We agree that MPI ETa dataset (50 km) is too coarse to capture the variability in ET at point scale. However, in this study, we used MPI ETa to validate SSEBop at basin/region scale. MPI ET was found to capture the signal in ET over basin scale efficiently [2,3]. Although we did not investigate GLDAS for this study, we will keep GLDAS in mind for future efforts, especially as part of a multi-model inter-comparison study.

5.     The mass balance approach should be used in Basin-scale validation of ET.

Authors response: We agree a water balance approach validation would be great but since we already presented point-scale validation using FLUXNET data and basin scale validation using MPI, we did not demonstrate water balance validation for this study. Besides, a water balance validation for a highly engineered-basin like URGB would require river diversion and groundwater abstraction data that are not easily available.  Discussion is underway for a separate study to understand the water budget dynamics of the basin.

6. How the missing values in remote sensing would have effects on the trend analysis?

Authors response: Usually, the Mann-Kendall test can be computed even if there are missing values. However, the performance of the test can be adversely affected by missing values. However, since we used peak seasonal (June-July-Aug) ET for the trend analysis, we do not have any missing values in our analysis.

Reviewer 2 Report

The authors use the SSEBop model to estimate evapotranspiration at 30m resolution over the Upper Rio Grande Basin (URGB) from 1986-2015.  SSEBop ET was compared with flux tower measurements and with a second, coarse-resolution (50km) ET product.  Trends in ET were quantified using Mann-Kendall trend tests by State.

Overall the paper is well written and the methods well-conceived and properly implemented (though I have grammatical and writing suggestions throughout).  I didn’t see the value in the comparison of trends in ETa with trends in NDVI over each sub-basin.  The authors should more clearly state why such an analysis is important, and what question is being addressed.  Much of the discussion is confined to listing the results by basin, which wasn’t terribly interesting as it wasn’t motivated by a question.  

Specific comments.

Figure 4 is difficult to interpret, since the two datasets are on separate panels.  At a minimum the Eta time series (3yr and cumulative averages) should be included on the other panel for comparison.  You write that the magnitudes are not directly comparable (line 434).   Resampling SSSEBop to have the same geographic coverage would be the best comparison…why not do that?  At least, the anomalies could be plotted together on one chart…?

Figure 10.  You conclude that more of the basin showed negative trends than showed positive trends.  Most of the negative trends are in the are outside the pivot crop circles.  Are those areas cultivated?  If not how would that change your interpretation?

L456.  I find the switching back and forth between Eta and “water use” unnecessarily confusing.  I know you’re taking about consumptive water use, but why not just stick with "ETa"?  Your audience will know the implications of ET.  L449 is fine since it specifies “water use from Eta over irrigated agriculture”.

L454, and in general: “which” is usually preceded by a “,”.

L466.  “This once again demonstrates 466 the water management response to environmental stressors”.  Did they reduce water consumption in response to environmental stressors alone, or did management/policy play a role?  In other parts of the western US, ET may not have fallen at all due to water rights.  Don’t your results show that they have low-priority water rights?

L639.  Table 4.  Why are there units on each variable…you just show the p-value and slope.  And, isn’t the units on the slope, for ETo for example, in mm/yr per year?

Author Response

Reviewer 2:

The authors use the SSEBop model to estimate evapotranspiration at 30m resolution over the Upper Rio Grande Basin (URGB) from 1986-2015.  SSEBop ET was compared with flux tower measurements and with a second, coarse-resolution (50km) ET product.  Trends in ET were quantified using Mann-Kendall trend tests by State.

Overall the paper is well written and the methods well-conceived and properly implemented (though I have grammatical and writing suggestions throughout).  I didn’t see the value in the comparison of trends in ETa with trends in NDVI over each sub-basin.  The authors should more clearly state why such an analysis is important, and what question is being addressed.  Much of the discussion is confined to listing the results by basin, which wasn’t terribly interesting as it wasn’t motivated by a question. 

Authors response:

The NDVI analysis is included as part of the set of agro-hydro/climatological explanatory variables.  We agree, we could improve the discussion how these agro-hydro-climatic variables are used in shedding more light in understanding the spatiotemporal dynamics of water use in relation to drivers (precipitation and ETo) and responses (ETa, LST, NDVI, runoff). We have revised the manuscript to address these weaknesses.

Specific comments.

Figure 4 is difficult to interpret, since the two datasets are on separate panels.  At a minimum the Eta time series (3yr and cumulative averages) should be included on the other panel for comparison.  You write that the magnitudes are not directly comparable (line 434).   Resampling SSSEBop to have the same geographic coverage would be the best comparison…why not do that?  At least, the anomalies could be plotted together on one chart…?

Author’s response:  Thank you for the suggestion. We modified the figure as per your suggestion by adding an additional panel to display the line graphs on the same chart. The suggestion to resample the SSEBop to same coarse resolution is valid and more accurate for comparing the absolute magnitudes. Because our interest was on looking at the trends of both datasets, we believe the conclusion would not change with existing sampling, but we will modify the discussion by pointing out the more correct approach to compare the two products for absolute magnitudes would be resampling the finer resolution to the coarser dataset.

Figure 10.  You conclude that more of the basin showed negative trends than showed positive trends.  Most of the negative trends are in the are outside the pivot crop circles.  Are those areas cultivated?  If not how would that change your interpretation?

Author’s response:

We believe we did not do good job expressing our interpretations on this. What we observed and concluded was that most of the area did not show any trends, i.e., was statistically neutral.  But for the parts that showed trends (less than 20% for the basin, less than 40% for croplands), there were more negative than positive trends, giving it an overall declining trend. So, our conclusion was that not every pixel is declining, but the trend comes from a small percentage of the study area. Discussion has been modified.

L456.  I find the switching back and forth between Eta and “water use” unnecessarily confusing.  I know you’re taking about consumptive water use, but why not just stick with "ETa"?  Your audience will know the implications of ET.  L449 is fine since it specifies “water use from Eta over irrigated agriculture”.

Author’s response:

We accept the criticism on this. We have revised the manuscript to have consistency when we use ET and water use. We will be using ETa in most instances (e.g., modeling and in relation to other variables). We refer to “water use” when referring to management relevant discussion in volumetric basis with sufficient reminders that is basically ETa. We modified all figures to be labeled as ETa.

L454, and in general: “which” is usually preceded by a “,”.

Author’s response:  Done. We modified the sentence accordingly.

L466.  “This once again demonstrates 466 the water management response to environmental stressors”.  Did they reduce water consumption in response to environmental stressors alone, or did management/policy play a role?  In other parts of the western US, ET may not have fallen at all due to water rights.  Don’t your results show that they have low-priority water rights?

Author’s response: We have revised the text to clarify this point. We were not trying to imply one or the other, which requires more investigation to control one of the two drivers (climate vs management). In this case, we were indicating that ET reduction was obvious in drought years and thus capturing the impact of climate signals. We agree, that the reduction in ET could be lack of available water due to declining resources, whose spatial impact will depend on water rights. Or it could also be due to changes in land and water management decisions. However, we don’t have information to separate the two sources other than pointing out the more obvious drought years and their impact on ET. We have revised the statement to read “This once again demonstrates the capability of remote sensing to monitor the impact of climatic and environmental stressors on water resources and landscape responses.”

L639.  Table 4.  Why are there units on each variable…you just show the p-value and slope.  And, isn’t the units on the slope, for ETo for example, in mm/yr per year?

Author’s response:  Thanks for pointing this out.  We have struggled both ways.  Although the units are straightforward, for the table to stand by itself and avoid confusion what units were used for each parameter, we decided to keep the units on the “parent” variable and not for the derived parameters such as slope, unless it was to indicate the time unit for the rate (e.g., units per decade).

Reviewer 3 Report

The present manuscript introduces an impressive work of data reconstruction based on Landsat data carried out on the  Upper Rio Grande Basin. The authors reconstructed the time series of Eta over the period 1986-2015. The dataset allowed to identify interesting trend in the data demonstrating a trend in the reduction of crop water use. I would like to strongly recommend this manuscript for publication. I have few comments for the authors with the scope to improve the final quality of the manuscript: 1) I feel that the manuscript is a bit too long. If you are able to summarize of remove some sections, I think that this will be beneficial to increase the readability of the manuscript. There is a bit of unbalance between the length of the manuscript and the number of results presented. I understand that there is a lot of work done, but some of it can be removed or moved in an appendix without losing much in the final message. 2) Calculation of ETo was carried out making some assumption not fully justified. What is the reason for changing the Ta only above 1500m a.s.l.? 3) In all sites there is a clear reduction of precipitation. Such reduction can impact significantly water use especially in water limited environment. Is it responsible of the observed reduction of ETa? 4) another potential aspect that may be investigated in the discussion is represented by the agronomic trends in irrigations system. In the last 30 years there have been an evolution in this technologies investing a lot on water saving systems. This element should be taken in to consideration in the study.

Author Response

Reviewer 3:

The present manuscript introduces an impressive work of data reconstruction based on Landsat data carried out on the Upper Rio Grande Basin. The authors reconstructed the time series of Eta over the period 1986-2015. The dataset allowed to identify interesting trend in the data demonstrating a trend in the reduction of crop water use. I would like to strongly recommend this manuscript for publication. I have few comments for the authors with the scope to improve the final quality of the manuscript:

1) I feel that the manuscript is a bit too long. If you are able to summarize of remove some sections, I think that this will be beneficial to increase the readability of the manuscript. There is a bit of unbalance between the length of the manuscript and the number of results presented. I understand that there is a lot of work done, but some of it can be removed or moved in an appendix without losing much in the final message.

Author’s response:

We appreciate your suggestion. We have removed redundant discussions that appeared both at the results section and discussion.  For example, several paragraphs were removed (or moved to discussion) from the results as they have been presented in the discussion with comparable effect.

2) Calculation of ETo was carried out making some assumption not fully justified. What is the reason for changing the Ta only above 1500m a.s.l.?

Author’s response: We think there is a misunderstanding on the use of Ta and its adjustment.  We would like to clarify that the ETo was downloaded from GRIDMET using the standardized P-M algorithm [4]. We did not modify the ETo. But we modified the daily maximum temperature (Ta) as used as a model parameter to define the wet/cold boundary limit.  The Ta was modified for elevations above 1,500 m based on earlier suggestions (Senay, 2018) to account for observed differences in lapse rate between LST and Ta. The lapse difference was more apparent and important above 1500 m (this was an “empirical threshold” from trial and error along several transects in the western US. Changing this threshold by +/- 200 would not change the result substantially. The most noticeable effect occurs above 2500, i.e., 1000 m above the threshold level with a potential temperature difference of 3 K (without adjustment). Note that the modified Ta was not used in the ETo calculation.

It is possible this threshold and rate of adjustment can vary from place to place and from datasets to dataset, i.e., a Ta derived from another source may not need any adjustment or require a different rate. As SSEBop is a blend of empirical and physics-based algorithms, we listed parameters used in the model so users know what was used in this version of the particular odel run and with a guidance to modify the parameter as needed.

3) In all sites there is a clear reduction of precipitation. Such reduction can impact significantly water use especially in water limited environment. Is it responsible of the observed reduction of ETa?

Author’s response: Yes. In the rainfed areas, all the reduction in ET can be directly attributed to the reduction in the precipitation. However, for cropland pixels, the reduction in precipitation may or may not result in decrease in ET directly and would depend on water management practices in the region. For regions where surface water is drawn directly from the river/stream, reduction in precipitation can directly lead to decline in river flow and thereby result in reduction in ET. In places where groundwater is abstracted for irrigation, precipitation decline may not result in decline in ET.

In dry regions of the southwest United States, reduction in water supply (due to reduction in precipitation) can lead to reduction in the cropped area and volumetric water use (California central valley study, Schauer and Senay, 2018, under review), but ET per unit area (mm/year) may remain the same.

4) another potential aspect that may be investigated in the discussion is represented by the agronomic trends in irrigations system. In the last 30 years there have been an evolution in this technology investing a lot on water saving systems. This element should be taken in to consideration in the study.

Author’s response: This is a good point. We added few sentences in the discussions. This study points out the overall changes in the ET over the URGB. We cannot pin point to what extent the changes in ETa are due to various drives such as precipitation, water and land management practices or climate change etc.  Discussion is underway for a separate study to understand the water budget dynamics of the basin, which could include the impact of changes in water management and irrigation technology.
